# Transport mechanisms of hydrothermal convection in faulted tight sandstones

Guoqiang Yan; Benjamin Busch; Robert Egert; Morteza Esmaeilpour; Kai Stricker; Thomas Kohl

Institute of Applied Geosciences, Karlsruhe Institute of Technology (KIT), Karlsruhe, 76131, Germany

*Correspondence to*: Guoqiang Yan (guoqiang.yan@kit.edu)

**Abstract.** Motivated by the unknown reasons for a kilometer-scale high-temperature overprint of 270 ~ 300 °C in a reservoir outcrop analog (Piesberg Quarry, northwestern Germany), numerical simulations are conducted to identify the transport mechanisms of the fault-related hydrothermal convection system. The system mainly consists of a main fault and a sandstone reservoir in which transfer faults are embedded. The results show that the buoyancy-driven convection in the main fault is the
basic requirement for elevated temperatures in the reservoir. We studied the effects of permeability variations and lateral regional flow mimicking the topography conditions on the preferential fluid flow pathways, dominant heat transfer types, and mutual interactions among different convective and advective flow modes. The sensitivity analysis of permeability variations indicates that lateral convection in the sandstone and advection in the transfer faults can efficiently transport fluid and heat, thus causing elevated temperatures ($\geq$ 269 °C) in the reservoir at the depth of 4.4 km compared to purely conduction-dominated
heat transfer ($\leq$ 250 °C). Higher-level lateral regional flow interacts with convection and advection and changes the dominant heat transfer from conduction to advection in the transfer faults for the low permeability cases of sandstone and main fault. Simulations with anisotropic permeabilities detailed the dependence of the onset of convection and advection in the reservoir on the spatial permeability distribution. The depth-dependent permeabilities of the main fault reduce the amount of energy transferred by buoyancy-driven convection. The increased heat and fluid flows resulting from the anisotropic main fault
permeability provide the most realistic explanation for the thermal anomalies in the reservoir. Our numerical models can facilitate exploration and exploitation workflows to develop positive thermal anomalies zones as geothermal reservoirs. These preliminary results will stimulate further petroleum and geothermal studies of fully coupled thermo-hydro-mechanical-chemical processes in faulted tight sandstones.

## 1    Introduction

Thermal anomalies in sedimentary basins are common geological phenomena and may affect hydrocarbon reservoir properties and the utilization of subsurface space concerning porosity and permeability due to cementation with temperature-related authigenic cement (Baillieux et al., 2013). Meanwhile, the thermal anomalies in sedimentary basins can be targeted as geothermal resources for heat and electricity production (Moeck, 2014). In general, thermal anomalies may be caused by variations of thermal conductivities around structures such as salt domes (O'Brien and Lerche, 1988; Magri et al., 2008),
geological/tectonic activity (Emry et al., 2020), geochemical reactions (Elderfield et al., 1999), or hydrothermal activities in faults and fractures (Cherubini et al., 2013). Indeed, in addition to being a possible cause of thermal anomalies (Zwingmann et al., 1998, 1999; Liewig and Clauer, 2000; Will et al., 2016), the hydrothermal activities in faults and fractures are often invoked to explain natural processes in sedimentary basins, such as hydrothermal mineralization (Kühn and Gessner, 2009; Harcouët‐menou et al., 2009). The common driving force for hydrothermal activities in faulted and fractured lithologies is
the fluid density difference caused by temperature variations (Nield and Bejan, 2017). The resulting process is called buoyancy-driven convection/flow. Many numerical models imply that buoyancy-driven convection in faulted basins could explain the correlation between faulted zones or adjacent lithologies and thermal anomalies (Bruhn et al., 1994; Andrews et al., 1996; Lampe and Person, 2000). Besides the buoyancy-driven force, pore pressure gradients or lateral regional flow (LRF) from

topography, magmatic, or metamorphic fluid production also cause hydrothermal activities (e.g. controlling the fluid flow and heat transfer) (Wisian and Blackwell, 2004b), while the two flow modes may also coexist (McKenna and Blackwell, 2004). Several studies further reveal that these flow modes generate complex flow patterns within the faults (i.e., convection cells) and/or across the adjacent permeable lithologies (Magri et al., 2016).

Transport mechanisms of hydrothermal convection in faulted zones, including preferential fluid flow pathways, dominant heat transfer types (i.e., conduction, advection, and convection), and the mutual interactions of different flow modes, are crucial for interpreting related diagenetic, petrophysical, and thermal phenomena in sedimentary basins. Therefore, it is necessary to understand the transport mechanisms of hydrothermal convection in faulted zones, which are mainly controlled by the permeability distributions of the fault systems and the surrounding lithologies, pore pressure gradients, and LRF (López and Smith, 1995; Fairley, 2009). The dominant heat transfer types are quantified by analyzing the Peclet number, which in this case represents the ratio of heat transfer by convection/advection to heat transfer by conduction (Jobmann and Clauser, 1994). For Peclet numbers much less than 1 (e.g., < 0.1), the system is conduction-dominated, while for values greater than 1 the system is convection/advection-dominated (Kämmlein et al., 2019). A Peclet number within the range of 0.1 and 1 indicates a convective-conductive system (Beck et al., 1989). Furthermore, rock heterogeneities (e.g. anisotropic and depth-dependent permeabilities), mainly resulting from sedimentary depositional environments, superimposed tectonics, and lithostatic stress, are common phenomena in sedimentary basins (Parry, 1998; Achtziger-Zupančič et al., 2017b; Panja et al., 2021). Affected by the preferential orientation of pore connectivity during deformation, the orientation of maximum permeability of faulted zones commonly lies parallel to the fault plane (i.e., parallel to bedding) (Scibek, 2020). The horizontal permeability of sedimentary rocks generally shows higher values than the vertical (i.e., perpendicular to bedding) permeability due to the effect of bedding, as controlled by the sedimentary depositional environments (Panja et al., 2021). Permeability is generally expected to decrease with increasing lithostatic pressure (Manning and Ingebritsen, 1999; Saar and Manga, 2004). Anisotropic and depth-dependent permeabilities may change the fluid flow pathways and affect heat transfer (Guillou-Frottier et al., 2020). López and Smith (1995) and Magri et al. (2016) presented 3D numerical investigations involving topography-driven flow and buoyancy-driven flow to infer the transport mechanisms of a hydrothermal convection system. Magri et al.' hydraulic conditions were isotropic and homogeneous, whereas López and Smith's simulation set partly anisotropic and heterogeneous hydraulic conditions. The effects of thermo-hydraulic conditions, including permeability distributions of fault systems and surrounding lithologies, and LRF, on the transport mechanisms, have not yet been systematically investigated.

A blueprint for complex hydrothermal conditions is the Piesberg Quarry, exposing Upper Carboniferous sandstones in the Lower Saxony Basin (LSB) in northwestern Germany. The quarry is chosen as a reservoir analog because of its similarity to still producing tight sandstone reservoirs in LSB concerning sedimentology, stratigraphy, fault patterns, and average porosity. Tight sandstone, generally referred to low permeability reservoirs that produce mainly dry natural gas, is a common play type of petroleum reservoir (Becker et al., 2019). The Upper Carboniferous sandstone reservoir analog, once located at a maximum depth of about 5 km, was affected by a kilometer-scale thermal anomaly (270 ~ 300 °C), as interpreted from geothermometry data including fluid inclusion, K/Ar-dating, chlorite thermometry, and vitrinite reflectance analyses (Wüstefeld et al., 2017b). The geothermometry data indicated that the chlorite in veins typically reaches temperatures of ~ 300 °C due to fluid flow, whereas the pore-filling chlorite records an average temperature of 270 °C. Furthermore, the Piesberg Quarry is also an experimental study case for the thermal control of a porosity-permeability evolution defined by long-term history matching in this play type (Becker et al., 2019). Thus, revealing the transport mechanisms of the hydrothermal convection system in the Piesberg Quarry is an effective strategy for interpreting related geological phenomena such as kilometer-scale thermal anomalies).

Our study aims to systematically identify the transport mechanisms as inferred from the results of fluid flow and heat transfer in the tight sandstone reservoir. For the first time, the example of the Upper Carboniferous Piesberg Quarry and its geological past is used to numerically investigate the physical processes leading to the kilometer-scale thermal anomaly in

faulted tight sandstones. A 3D model is established based on the geological conditions of the Piesberg Quarry. Firstly, considering complex thermo-hydraulic conditions, numerical modeling is performed, and the influence of the different transport mechanisms is quantified. Then, the transport mechanisms are further assessed and discussed to study the effects of the anisotropic and depth-dependent permeabilities on the transport mechanisms. Possible reasons for the kilometer-scale thermal anomaly in the Piesberg Quarry are inferred. Based on the generalized structural characteristics of the studied domain, the numerical results can be applied to interpret hydrothermal convection-related geological phenomena and to draw implications for future exploration and exploitation of reservoirs in analogous settings.

## 2     Material and Methods

### 2.1     The reservoir analog study area

The studied Piesberg Quarry is located at the southwestern margin of the LSB (Figs. 1a and 1b). The reservoir-scale Piesberg study site has dimensions of approximately 1 km in the W-E direction and 0.5 km in the N-S direction, with a depth of roughly 0.1 km (Fig. 1c).

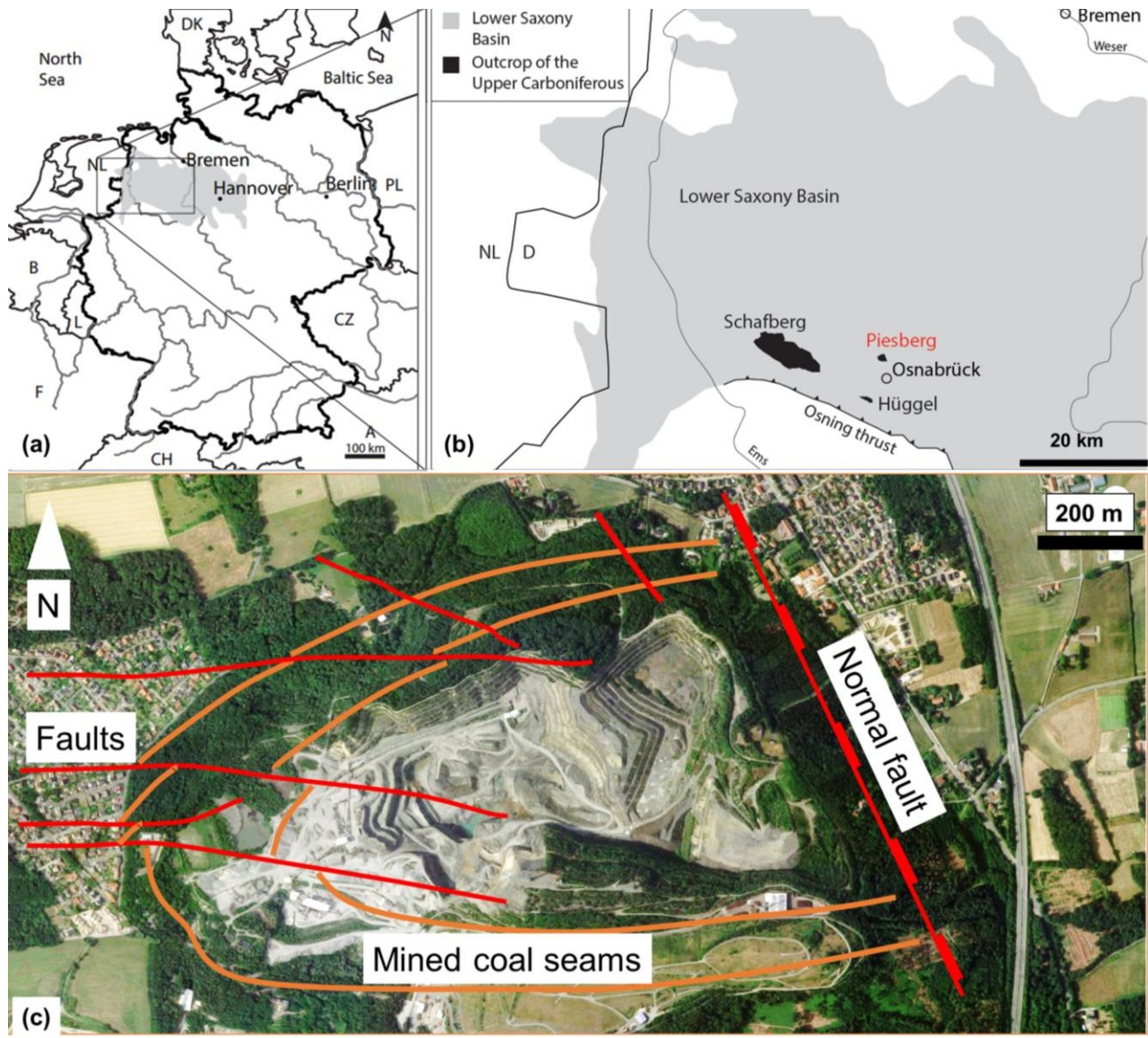

**Figure 1.** (a) Location of the studied area in northwestern Germany. (b) Detailed location of the investigated Piesberg Quarry at the southern rim of the LSB (redrawn from Senglaub et al. (2006) and Busch et al. (2019)). (c) Aerial image of the Piesberg Quarry (© Google Maps, 2023). The coal seams, faults, and the main normal fault in the east are based on historical subsurface mining data (Haarmann, 1909; Wüstefeld et al., 2017b).

Within the Piesberg Quarry, two dominant fault systems can be distinguished (Fig. 1c). First, the NNW-SSE striking normal fault (at ~ 170° strike angle) is represented in the east of the quarry (not exposed anymore), with a down-dip displacement of up to 600 m (Hinze, 1979; Baldschuhn R. et al., 2001). The second main strike orientation of the normal faults is W-E (at ~ 95° strike angle) to WNW-ESE, with average fault displacements of 10 m (Wuestefeld et al., 2014; Wüstefeld et al., 2017a; Becker et al., 2019). The initial total thickness of Upper Carboniferous sandstones in the region was 1.5 km (David et al., 1990). During the burial, the thickness of the sandstones decreased to nearly 1 km in the Late Jurassic to Early Cretaceous times. The sandstones are overlain by impermeable claystone and Zechstein cap rocks (Becker et al., 2019).

The geothermometry data from the outcrop samples in the Piesberg Quarry show a high-temperature thermal overprint of about 270 ~ 300 °C at about 4.4 km depth around the significant NNW-SSE striking normal fault at about 160 Ma. The NNW-SSE striking normal fault may thus have acted as a conduit for the heat/fluid source of the thermal overprint. Wüstefeld et al. (2017b) concluded that the local thermal increase of approximately 90 ~ 120 °C was the result of hydrothermal fluids circulating along the fault damage zone of the NNW-SSE striking normal fault, which laterally heated the tight sandstones. Furthermore, it is noteworthy that the W-E striking faults within the tight sandstones may also affect fluid flow and heat transfer processes because of their higher permeability (i.e., $10^{-13}$ m$^2$) and intersection with the NNW-SSE striking fault. Even though there are no field observations and evidence, LRF resulting from the topography conditions are hypothesized to investigate the effect of topography conditions on the transport mechanisms of hydrothermal convection. Considering the geological conditions, especially fault patterns and stratigraphy, the Piesberg Quarry area might be an example of the coexistence of buoyancy-driven convective and lateral regional flows. Thermo-hydraulic numerical simulations are required to investigate and quantify the plausibility of different hypotheses of transport mechanisms that could contribute to the kilometer-scale thermal anomaly.

## 2.2 Numerical workflow

### 2.2.1 Governing equations

Inspired by the Piesberg Quarry, an idealized and synthetic numerical analog is created to characterize geological conditions in faulted tight sandstones and to enable investigation of the transport mechanisms leading to fluid flow and heat transfer. Only single-phase liquid flow is considered in this study. The numerical simulations for coupled fluid flow and heat transfer processes are carried out with a finite element open-source application called TIGER (THMC sImulator for GEoscience Research) (Gholami Korzani et al., 2019; Egert et al., 2020; Egert et al., 2021), which is based on the MOOSE (Multiphysics Object-Oriented Simulation Environment) framework (Permann et al., 2020). TIGER has been developed to tackle thermo-hydraulic-mechanical-chemical problems in petroleum and geothermal reservoirs. The numerical study of thermal convection in permeable and porous media involves coupling fluid flow and heat transfer equations as shown below (Gholami Korzani et al., 2020). The mass transport equation is written as

$$S_m \frac{\partial p}{\partial t} + \nabla \cdot \boldsymbol{q} = 0 \tag{1}$$

where the subscript $m$ means porous media, $S_m$ is the constrained specific storage of the porous media (Pa$^{-1}$), $p$ is the pore pressure (Pa), $t$ is time (s), and $\boldsymbol{q}$ is the Darcy velocity in the porous media (m·s$^{-1}$). The $S_m$ comprised a mechanical alteration in response to pressure and is assumed as a constant given by $S_m = (1 - n)/K^s + n/K^l$ with porosity $n$ (-), the bulk moduli of solid $K^s$ and of liquid $K^l$ (Pa) (Watanabe et al., 2017). The Darcy velocity is given as

$$\boldsymbol{q} = \frac{k}{\mu}(-\nabla p + \rho^l \boldsymbol{g}) \tag{2}$$

where $k$ is the permeability (m$^2$), $\mu$ is the fluid dynamic fluid viscosity (Pa s), the superscript $l$ means liquid phase, $\rho^l$ is the fluid density (kg·m$^{-3}$), and $\boldsymbol{g}$ is the gravitational acceleration vector (m·s$^{-2}$).

The heat transfer equation is written as

$$\left[nc_p^l\rho^l + (1-n)c_p^s\rho^s\right]\frac{\partial T}{\partial t} - \left[n\lambda^l + (1-n)\lambda^s\right]\nabla^2 T + \rho^l c_p^l \boldsymbol{q}\nabla T - Q = 0 \tag{3}$$

where the superscript $s$ means solid phase, $c_p^l$ and $c_p^s$ are the specific heat capacity of the liquid and solid phases (J·m⁻³·K⁻¹), $\rho^s$ is the solid density (kg·m⁻³), $T$ is the temperature (K), $t$ is the time (s), $\lambda^l$ and $\lambda^s$ are the heat conductivity of the liquid and solid phases (W·m⁻¹·K⁻¹), $\boldsymbol{q}$ is the Darcy velocity in the porous media (m·s⁻¹), and $Q$ is the heat source (W·m⁻³).

### 2.2.2 Numerical model

The numerical model represents the stratigraphic distributions of the Piesberg Quarry region (belonging to the sandstone reservoir) at about 160 Ma. The model geometry extends 5 km along the W-E direction (i.e., X-axis), 12 km along the N-S direction (i.e., Y-axis), and over a depth of 12 km (i.e., Z-axis). The NNW-SSE striking fault ("main fault" in Fig. 2a) has a damage zone width of 400 m based on damage zone scaling relations to the fault displacement (Shipton et al., 2006; Torabi and Berg, 2011), a length of 12 km, and a height of 7 km (ranging from -11 km to -4 km on the Z-axis (Schultz et al., 2006)). In addition, three W-E striking faults ("transfer faults") with 10 m width (along Y-axis) and 100 m height (along Z-axis) are embedded in the sandstone (Fig. 2b). To the west of the main fault, the sandstone is located at 3.9 ∼ 4.9 km depth; to the east of the main fault, it is located at 4.5 ∼ 5.5 km depth due to the 600 m displacement of the main fault. Lithologies that have similar permeability and heat conductivity are merged into a single lithological unit. Basement layers mainly consist of shale, limestone, and sandstone. The claystone and Zechstein are lumped into caprock at the top of the model. The model consists of 174,675 nodes connecting 827,610 tetrahedra, with mesh sizes of 1000 m for low permeability zones and mesh sizes of 100 m for high permeability zones. Preliminary mesh sensitivity analyses were performed to avoid any mesh dependency on the results.

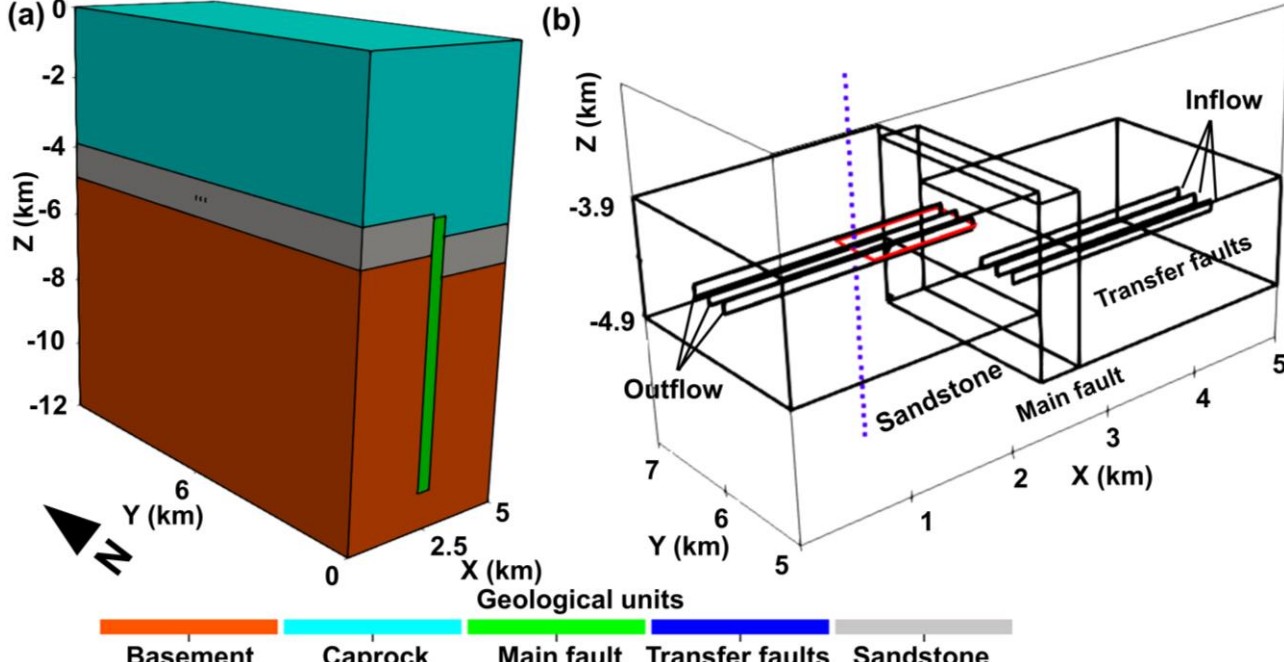

**Figure 2.** Model geometry of (a) the total model; (b) zoomed part of the sandstone reservoir where the transfer faults are embedded. The red rectangle, which is perpendicular to the main fault, represents the outline of the Piesberg Quarry. The dashed purple line shown in (b) is used to illustrate the extent of the temperature-depth profile.

The physical properties of each lithological unit are summarized in Table 1. The typical parameterization of main fault permeability ($\kappa_{MF}$) and sandstone permeability ($\kappa_{SST}$) are $10^{-13}$ m² and $10^{-15}$ m², respectively. The transfer faults permeability ($\kappa_{TF}$) is constant at $10^{-13}$ m².

**Table 1.** Parameters of the lithological units (Bruns et al., 2013; Wüstefeld et al., 2017a; Becker et al., 2019; Busch et al., 2019).

| Properties | Symbols | Units | Caprock | Main Fault | Sandstone | Transfer faults | Basement |
|---|---|---|---|---|---|---|---|
| Porosity | $n$ | - | 0.01 | 0.26 | 0.07 | 0.26 | 0.01 |
| Bulk modulus | $K_s$ | Pa | $10^{10}$ | $10^{10}$ | $10^{10}$ | $10^{10}$ | $10^{10}$ |
| Permeability | $k$ | $m^2$ | $5 \cdot 10^{-18}$ | $10^{-15} \sim 10^{-13}$ | $10^{-16} \sim 10^{-14}$ | $10^{-13}$ | $10^{-20}$ |
| Density | $\rho^s$ | kg m$^{-3}$ | 1800 | 2000 | 2642 | 2000 | 2650 |
| Specific heat capacity | $c_p^s$ | J kg$^{-1}$ °K$^{-1}$ | 1000 | 1000 | 1000 | 1000 | 1000 |
| Heat conductivity | $\lambda^s$ | W m$^{-1}$ °K$^{-1}$ | 3 | 2 | 2.5 | 2 | 2 |

*The main fault and sandstone permeabilities are abbreviated as $\kappa_{MF}$ and $\kappa_{SST}$, respectively. The subscript MF means the main fault and the subscript SST represents sandstone.

Fluid properties are summarized in Table 2. The fluid density is a temperature/pore pressure-dependent variable. The following polynomial function fitting experimental data for pure water in the liquid phase is used to compute fluid density over temperature and pore pressure ranges of $273.15 \sim 1273.15$ K and $0 \sim 500$ MPa (Linstrom and Mallard, 2001).

$$\rho^l = 1006 - 3.922 \cdot 10^{-1} \cdot (T - 273.15) - 3.774 \cdot 10^{-3} \cdot (T - 273.15)^2 + 2.955 \cdot 10^{-6} \cdot (T - 273.15)^3 + 7.424 \cdot 10^{-7} \cdot p + 4.547 \cdot 10^{-9} \cdot p \cdot (T - 273.15) + 5.698 \cdot 10^{-11} \cdot (T - 273.15)^4 + 7.485 \cdot 10^{-12} \cdot p \cdot (T - 273.15)^2 - 4.441 \cdot 10^{-15} \cdot p^2 + 7.372 \cdot 10^{-15} \cdot p \cdot (T - 273.15)^3 - 1.793 \cdot 10^{-17} \cdot P^2 \cdot (T - 273.15) + 4.018 \cdot 10^{-21} \cdot p^2 \cdot (T - 273.15)^2 + 1.451 \cdot 10^{-23} \cdot p^3 + 1.361 \cdot 10^{-26} \cdot p^3 \cdot (T - 273.15) - 1.463 \cdot 10^{-32} \cdot p^4 \tag{4}$$

The dynamic viscosity of the fluid is temperature-dependent and follows the equation (Smith and Chapman (1983)):

$$\mu = 2.414 \cdot 10^{\left(\frac{247.8}{T - 144.15} - 5\right)} \tag{5}$$

**Table 2.** Fluid properties (Smith and Chapman, 1983; Linstrom and Mallard, 2001; Watanabe et al., 2017)

| Properties | Symbols | Units | Value |
|---|---|---|---|
| Coefficient of thermal expansion | $\beta$ | K$^{-1}$ | $2.14 \cdot 10^{-4}$ |
| Bulk modulus | $K_l$ | Pa | $2 \cdot 10^9$ |
| Density | $\rho^l$ | kg m$^{-3}$ | Equation (4) |
| Viscosity | $\mu$ | Pa s | Equation (5) |
| Specific heat capacity | $c_p^l$ | J kg$^{-1}$ K$^{-1}$ | 4194 |
| Heat conductivity | $\lambda^l$ | W m$^{-1}$ K$^{-1}$ | 0.6 |

The boundary and initial conditions are set as follows. An air surface pore pressure of $10^5$ Pa is fixed at the top of the model as a Dirichlet boundary condition. The initial condition of the hydraulic field corresponds to a hydrostatic gradient. The lateral sides of the model are assigned to no-flow boundary conditions in terms of pore pressure and temperature. The temperature at the top of the model is fixed at 20 °C as a Dirichlet boundary condition. A basal heat flux (BHF) of 0.1 W·m$^{-2}$ represents the sum of the mantle heat flow and the heat emitted by the decay of radioactive elements in the crust (Wisian and Blackwell, 2004b), and is set at the bottom of the model as a Neumann boundary condition. The International Heat Flow database (Lucazeau, 2019) showed two available surface heat flow data of 0.58 W·m$^{-2}$ and 0.84 W·m$^{-2}$ near the Piesberg Quarry. These data were measured at 60 km southwest and 42 km north of the Piesberg Quarry, respectively. The preliminary tests reveal that a BHF of 0.1 W·m$^{-2}$ is close to the minimum requirement for reproducing the kilometer-scale thermal anomaly of $270 \sim 300$ °C in the Piesberg Quarry. A typical threshold of BHF of 0.1 W·m$^{-2}$ for geothermal development is assumed here (e.g., Blackwell et al., 2006; Cloetingh et al., 2010; Erkan, 2015) even though there is no evidence for the high BHF of 0.1 W·m$^{-2}$ near the Piesberg Quarry areas during the Late Jurassic to Early Cretaceous times and at the current time. It must be noted that the magnitudes of these physical characteristics are not significant because the transport mechanism of hydrothermal convection is mostly controlled by permeability distributions of the main fault and sandstone, which can vary by many orders of magnitude. The initial condition of the thermal field is derived from the steady-state simulation of pure heat conduction to avoid the effects of initial temperature perturbation on mass transport and heat transfer. The solution of the initial condition of temperature in our model is shown as follows:

$$T_{\text{initial}}(z) = T_{\text{top}} + \left(\frac{Q_{\text{BHF}}}{n\lambda^l + (1-n)\lambda^s}\right) \cdot z \tag{6}$$

where $T_{\text{initial}}(z)$ is the initial temperature at $z$ (m) position, $T_{\text{top}}$ is the imposed surface temperature (°C), and $Q_{\text{BHF}}$ is the imposed BHF (W·m⁻²).

Under the coexistence of buoyancy-driven convection, lateral convection, and LRF, Neumann boundary conditions are superimposed on the eastern (as "inflow zones") and western (as "outflow zones") boundaries of the transfer faults, respectively (Fig. 2b). The transient simulations are run up to 1 Myr, which allows the simulations to reach an approximate thermo-hydraulic equilibrium to compare the results of different transport mechanisms of hydrothermal convection.

### 2.3    Simulation cases

Based on the permeability distributions in the reservoir and LRF boundary conditions, the following simulations have been carried out to identify the different fluid flow pathways (Fig. 3) and to understand how the reservoir parametrization impacts fluid flow and heat transport. Rayleigh number ($Ra$) is a dimensionless number to characterize the fluid's flow regime and is defined as the ratio of buoyancy and viscosity forces multiplied by the ratio of momentum and thermal diffusivities (e.g. Nield and Bejan, 2006):

$$Ra = \frac{k(\rho^l)^2 c_P^l g\beta\Delta T H}{\mu[n\lambda^l + (1-n)\lambda^s]} \tag{7}$$

where $\beta$ is the coefficient of fluid thermal expansion (K⁻¹), $\Delta T$ is the temperature variation (K) over the porous media height $H$ (m). The initial temperature and pore pressure of the main fault range from 152.5 °C to 502.3 °C and 39.3 MPa to 108 MPa over its 7 km height, respectively. Thus, according to Equation (4), the density and viscosity of the fluid are 921.5 kg m⁻³ and 1.8 10⁻⁴ Pa s at the lowest temperature and pore pressure conditions leading to

$$Ra_{\text{MF}} = \frac{10^{-15} \cdot (921.5)^2 \cdot 4200 \cdot 9.81 \cdot 2.14 \cdot 10^{-4} \cdot (502.3-152.5) \cdot 7000}{1.8 \cdot 10^{-4} \cdot [0.263 \cdot 0.6 + (1-0.263) \cdot 2]} \approx 63$$

for the lowest permeability (i.e., 10⁻¹⁵ m²) of the main fault in Fig. 2a. The $Ra$ is proportional to the permeability of the porous media. Buoyancy-driven convection in the main fault (Fig. 3, red dashed ovals) is triggered when its $Ra$ exceeds the critical Rayleigh number ($Ra^{crit}$). Furthermore, an analytical solution to calculate $Ra^{crit}$ values for the case of a porous media with dynamic fluid viscosity are shown as follows (Malkovsky and Magri, 2016):

$$Ra^{crit} = 0.25\left[\left(\frac{6.428}{\Delta}\right)^{1.165} + (27.1)^{1.165}\right]^{0.8584} \tag{8}$$

with half of the aspect ratio $\Delta = \frac{d}{2H} < 0.1$ where $d$ is the thickness of the porous media. In this studied case (Fig. 2a), the half aspect ratio of the main fault $\Delta_{MF} \approx 400/(2\ 7000) = 2.9\ 10^{-2}$ leads to the $Ra_{MF}^{crit} \approx 60$ for a temperature-dependent viscous fluid. This estimation suggests that the buoyancy-driven convection likely develops within the main fault even at the lowest permeability and temperature conditions. Furthermore, the presented model on the scale of faulted hydrothermal systems has been successfully validated against the results of the study conducted by Malkovsky and Magri (2016) and Guillou-Frottier et al. (2020) (shown in the Appendix). Lateral convection in the sandstone (Fig. 3, yellow arrows) can be enhanced if the pore pressure difference between the sandstone and the main fault decreases (McKenna and Blackwell, 2004). Natural advective flow in the transfer faults (Fig. 3, blue vectors) is getting significant when the transfer fault is more permeable than the main fault. Apart from the natural advective flow, forced advection in the transfer faults (Fig. 3, green vectors) gets significant if an LRF boundary condition is applied.

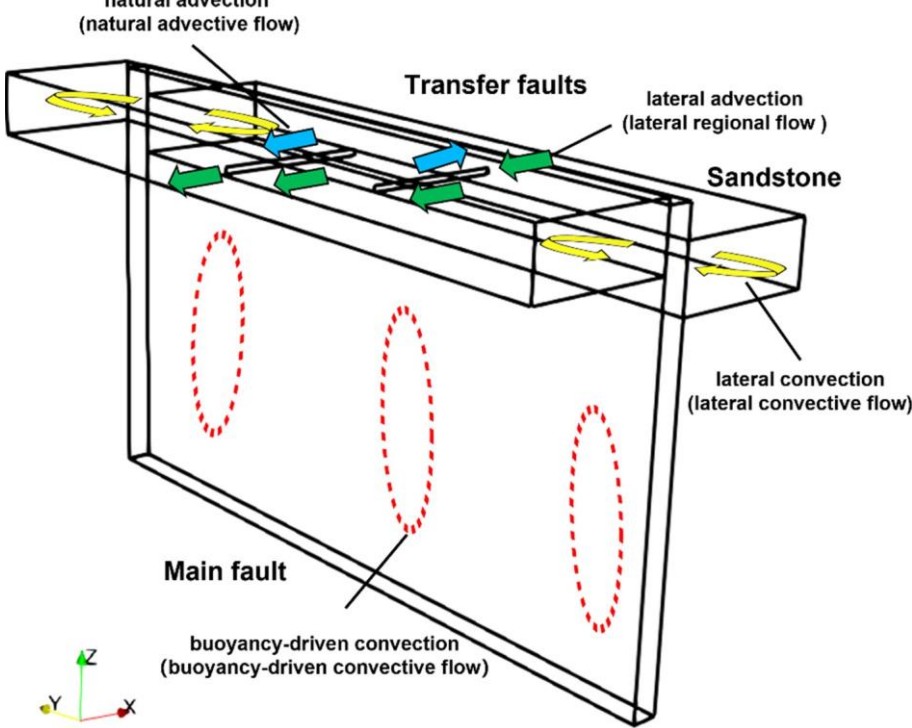

**Figure 3.** Schematic representation of the modeled 3D main fault. Four different types of fluid flow pathways involving the lithological units are illustrated. The 3D main fault account for the relative displacement (i.e., 600 m) of the surrounding lithological units. Only one transfer fault is shown on both sides of the main fault for simplicity.

Herein, a reference case with typical play type parameterization (Table 1) has been selected to demonstrate the general behavior of fluid flow pathways and heat transfer types in the hydrothermal convection system. Then a sensitivity analysis is

conducted on the $\kappa_{MF}$ (ranging from $10^{-15}$ m$^2$ to $10^{-13}$ m$^2$) and $\kappa_{SST}$ (between $10^{-16}$ m$^2$ and $10^{-14}$ m$^2$) to identify their effects on the preferential fluid flow pathways and dominant heat transfer types. Then, these above-described permeability cases are superimposed by three representative LRFs. The LRFs range from $10^{-8}$ m·s$^{-1}$ to $5 \times 10^{-7}$ m·s$^{-1}$ (LRF$_{max}$). Besides the preferential fluid flow pathways and heat transfer types, the mutual interactions between the LRF and convective/advective flows are studied.

Furthermore, the effects of anisotropic and depth-dependent permeabilities on the transport mechanism are discussed. Many researchers show that the degree of permeability heterogeneity greatly varies even within the meter scale of the lithological unit (Ingebritsen and Manning, 1999; Saar and Manga, 2004). However, our discussion only focuses on the effects of the anisotropic and depth-dependent permeabilities of the main fault on the transport mechanisms. This is because the variation in lithostatic stress and sandstone properties is smaller than that of the main fault, and the low value of median $\kappa_{SST}$

(i.e. $10^{-15}$ m$^2$). Farrell et al. (2014) shown the degree of anisotropy (i.e., horizontal permeability ($\kappa_{MF\_vertical}$)/vertical permeability ($\kappa_{MF\_horzontial}$)) of the fault ranges from one to three orders of magnitude. Thus, the $\kappa_{MF\_horizontal}$ is set at one to three magnitudes (i.e., $10^{-16} \sim 10^{-14}$ m$^2$) lower than $\kappa_{MF\_vertical}$ (i.e., $10^{-13}$ m$^2$), in which the anisotropic $\kappa_{MF}$ is investigated. Influenced by the in situ lithostatic pressure distribution in the LSB (Manning and Ingebritsen, 1999; Saar and Manga, 2004), the $\kappa_{MF}$ is exponentially decreased by one to three orders of magnitudes from its top locations (i.e., $10^{-13}$ m$^2$ at z = -4 km location) to its

bottom locations (i.e., $10^{-16} \sim 10^{-14}$ m$^2$ at z = - 11 km location). The above anisotropic and depth-dependent variations lie in the inferred ranges by Kuang and Jiao (2014) and Achtziger-Zupančič et al. (2017a).

In this study, the "thermal anomaly" is defined by the computed temperature minus the initial temperature distribution conditions. The maximum temperature at the western boundary of the Piesberg Quarry (T$_{max}$) is employed to compare with the geothermometer data. The temperature-depth profile following the western boundary of the Piesberg Quarry (Fig. 2b) and

the Peclet number of the lithological units are used to reveal the temperature distribution with depth and to identify the

dominant heat transfer types, respectively. The Peclet number is calculated as the ratio of the heat flow rate by convection to the heat flow rate by conduction for a uniform temperature gradient (Jobmann and Clauser, 1994), as follows:

$$Pe = \frac{\rho^l c_p^l q L}{n\lambda^l + (1-n)\lambda^s} \qquad (9)$$

where Pe is the Peclet number (-) and $L$ is the length scale of the fluid flow (m). The Peclet numbers are not uniform across

lithological units, therefore the statistically median value of the Peclet numbers rather than the Peclet number at a specific location (e.g., geometric center) is used to assess the general heat transfer types within the lithological units. The statistically median Peclet number of the main fault, transfer faults, and sandstone are abbreviated to Pe_MF, Pe_TF, and Pe_SST, respectively.

## 3    Results

### 3.1    Reference case

A reference case is developed to demonstrate the general behavior of fluid flow pathways and heat transfer types in the hydrothermal convection system, as shown in Fig. 4. In the initial state ($t = 0$ sec), the temperature and its gradient increase with depth depending on the thermal conductivities of deep lithological units (black line in Fig. 4a). At 1 Myr, the buoyancy-driven convection in the main fault with Pe_MF = 9.0 results in thirteen convective cells which consist of seven downflows and upflows in each at the maximum Darcy velocity of $1.1 \cdot 10^{-7}$ m·s$^{-1}$ (Fig. 4b). The buoyancy-driven convection is triggered by

fluid density variations resulting from the temperature difference. The mentioned upward and downward flows of fluids generate six full and one-half thermal plumes, as shown in Fig. 4c. The convective flow transports heat upwards inside the main fault. Thus, compared to the initial temperature distribution (black line in Fig. 4a), the buoyancy-driven convection results in positive thermal anomalies ranging between 0 °C and 86 °C in the shallow part of the main fault (i.e., z = - 6 ~ - 4 km locations in Fig. 4d).

Initiated by the same permeability of the main fault and transfer faults (i.e., $10^{-13}$ m$^2$), the lateral convection in the transfer faults with Pe_TF = 0.5 includes fluid recharge and discharge processes at the maximum Darcy velocity of $4.6 \cdot 10^{-8}$ m·s$^{-1}$ as shown in Fig. 4b. The heated fluid recharges from the main fault into the shallow part of the transfer faults (i.e., z = - 4.4 ~ - 4.35 km locations) and discharges from the deep part of the transfer faults (i.e., z = - 4.45 ~ - 4.4 km locations) into the main fault, respectively. However, the temperature-depth profile of the reference case (i.e., the red line in Fig. 4a)

shows that the lateral convection has a negligible effect on increasing the temperature in the transfer faults because of the narrow (i.e., width 10 m) and thin (i.e., thickness 100 m) geometry conditions of the transfer faults and the low permeability of the surrounding sandstone (i.e., $10^{-15}$ m$^2$).

Figure 4b shows that the reference case yields weak fluid flow with Darcy velocity $\leq 5 \cdot 10^{-9}$ m·s$^{-1}$ in the sandstone, caprock, and basement units ("off-faults domains") owing to their relatively low permeabilities (i.e., $\leq 10^{-15}$ m$^2$). The

temperature distribution in the off-fault domains (e.g., Pe < 0.1) is mainly dominated by heat conduction, as shown in Figs. 4a and 4c. Influenced by the thermal anomalies inside the main fault, heat conduction leads to upward-convex isotherms in the units located near the shallow region of the main fault (i.e., z = - 7.5 ~ - 4 km locations in Fig. 4c). However, heat conduction causes downward-convex isotherms in the units located near the deep region of the main fault (i.e., z = - 11 ~ - 7.5 km locations in Fig. 4c). The resulting upward- and downward-convex isotherms agree with an increase and decrease in

temperature above and below 7 km depth (red line in Fig. 4a) compared to the initial temperature profile (black line in Fig. 4a), respectively. Additionally, impacted by the convective mass flow, a pore pressure perturbation (~ -2.8 MPa) around the main fault extends ~ 0.3 km in the sandstone and transfer faults at a depth of 4.4 km.

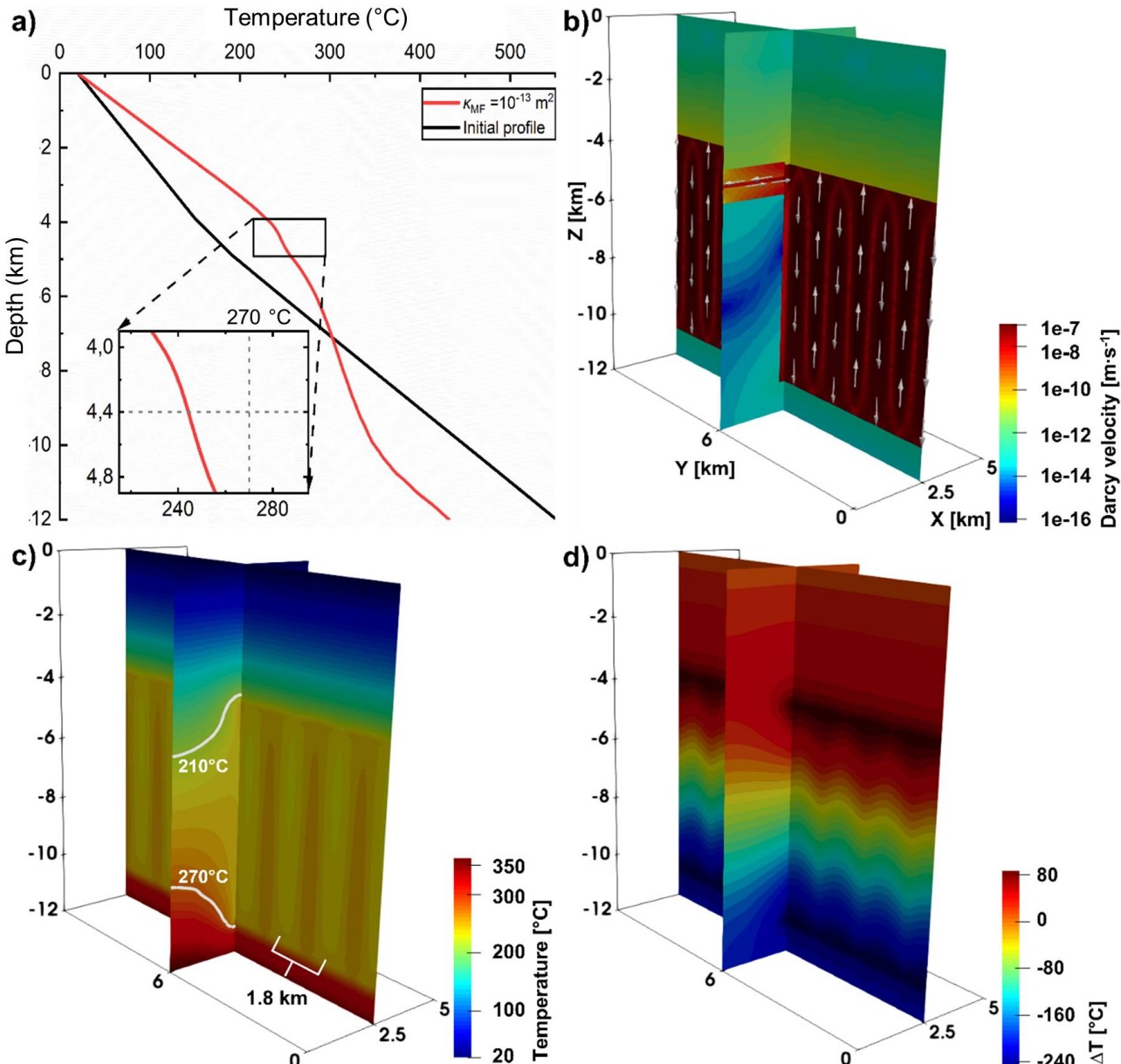

**Figure 4.** Fluid flow and temperature fields of the reference case. (a) Temperature-depth profiles that follow the western boundary of the Piesberg Quarry (Fig. 2b) at the initial time and when $T_{max}$ is reached, respectively. At 1 Myr, cross-sectional views of the (b) flow field with arrows representing the preferential fluid flow pathways; (c) resulting temperature distributions. The 210 °C and 270 °C isotherms are highlighted to indicate the dominant heat transfer types; (d) temperature differences compared to the initial conductive temperature distribution.

## 3.2 Effects of permeability distributions

### 3.2.1 Main fault

We evaluate how the $\kappa_{MF}$ variations affect the fluid flow pathways and heat transfer types in the hydrothermal convection system. Their differences compared to the reference case are described as follows. The $\kappa_{MF} = 10^{-14}$ m² case results in eight buoyancy-convection cells in the main fault with Pe_MF = 2.6 and three times lower maximum Darcy velocity of the convective flow than in the reference case with $\kappa_{MF} = 10^{-13}$ m². In addition to the above differences, the $\kappa_{MF} = 10^{-14}$ m² case has a similar focused flow in the faults and conduction-dominated heat transfer in the sandstone to the reference case. Results of fluid flow, temperature, and pore pressure change 1 Myr are illustrated in Fig. 5. for the $\kappa_{MF} = 10^{-15}$ m² case. It is observed in Fig. 5a that the $\kappa_{MF} = 10^{-15}$ m² case leads to two buoyancy-driven convection cells in the main fault with Pe_MF = 0.7 and eight times lower maximum Darcy velocity of the convective flow than in the reference case. Figures 5b and 5c show that the temperature increases in the center of the main fault ($\sim$ y = 6 km), associated with the upward flow of fluid, and decreases along the main fault's lateral sides ($\sim$ y = 0 and 12 km).

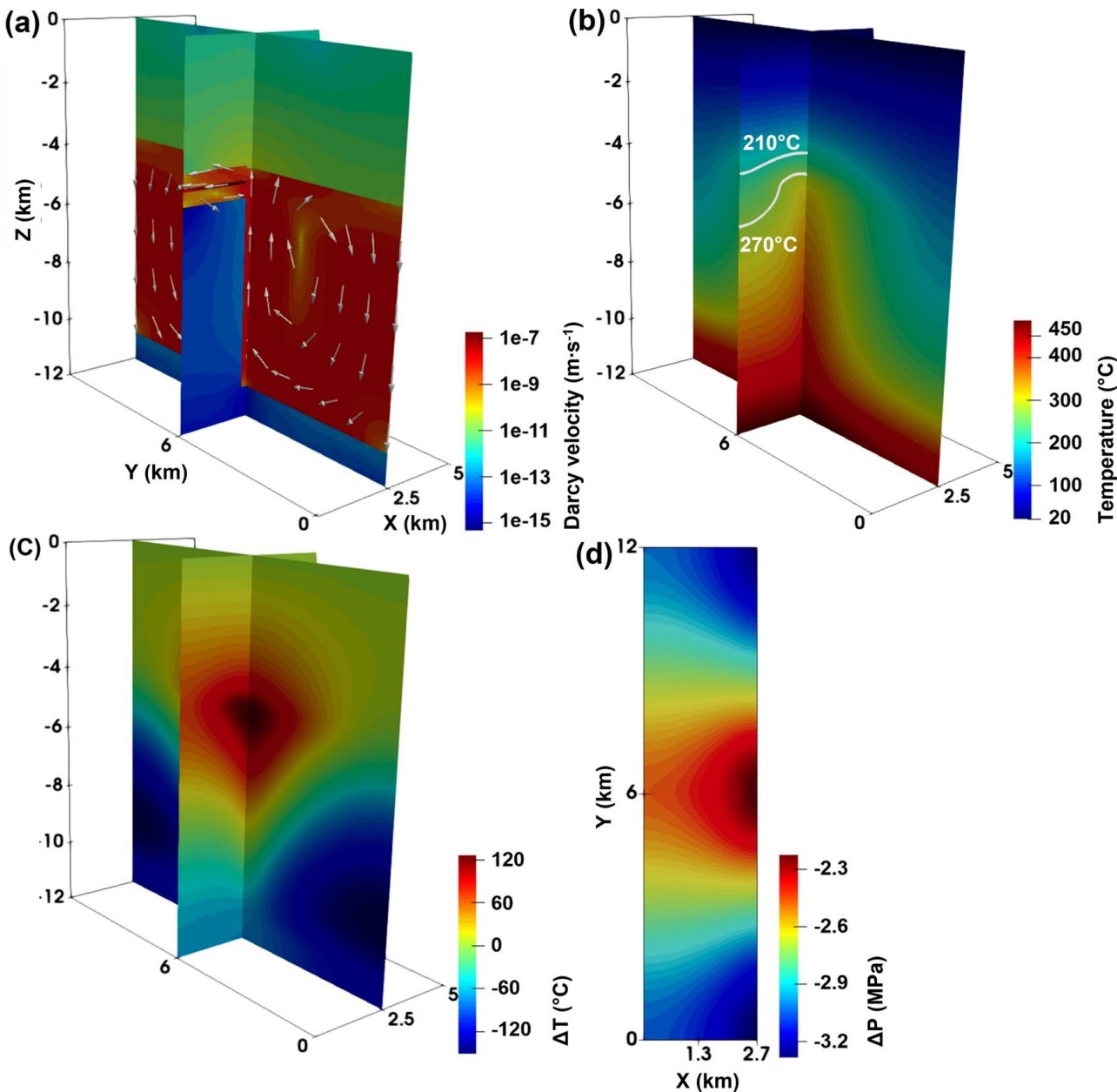

**Figure 5.** Fluid flow, temperature, and pore pressure change fields of the $\kappa_{MF} = 10^{-15}$ m$^2$ case at 1 Myr. Cross-sectional view of the (a) flow field with arrows representing the preferential fluid flow pathways; (b) resulting temperature field. The 210 °C and 270 °C are highlighted to indicate the dominant heat transfer types; (c) temperature differences compared to the initial conductive temperature distribution; (d) top view of the pore pressure change compared to the initial hydrostatic pore pressure (43.3 MPa) in the western sandstone, transfer faults, and main fault at a depth of 4.4 km. The 1.3 km and 2.7 km along the X-axis follow the west boundary of the Piesberg Quarry and the eastern boundary of the main fault, respectively.

From Fig. 5a, it is observed that the lateral convection can be enhanced in the sandstone, especially in the part (i.e., Pe_SST ≥ 0.9) near the main fault, by the equal $\kappa_{MF}$ and $\kappa_{SST}$ (i.e., 10$^{-15}$ m$^2$). The lateral convection consists of fluid recharge and discharge processes at a maximum Darcy velocity of 10$^{-8}$ m·s$^{-1}$, slightly lower than the buoyancy-driven convection. The heated fluid recharges from the central part of the main fault into the sandstone and discharges from the sandstone into the main fault's margins, respectively. Furthermore, caused by the large permeability contrast between the transfer faults and the main fault (i.e., $\kappa_{TF}/\kappa_{MF} = 100$), the natural advective flow in the transfer faults with Pe_TF = 3.9 propagates towards the lateral boundaries of the model as shown in Fig. 5a. The maximum Darcy velocity of the natural advective flow reaches 1.7·10$^{-7}$ m·s$^{-1}$, which is one order higher than the buoyancy-driven convection. Figures 5b and 5c demonstrate that, compared to the reference case, the increased mass flow involved in the lateral convection and natural advection effectively causes elevated temperatures in the sandstone and transfer faults near the center of the main fault. Affected by the increased mass flow, the

convective/advective flow processes in the main fault, sandstone, and transfer faults result in a larger ($\sim$ -3.2 MPa) and more extensive (0 $\sim$ 5 km) pore pressure perturbation than in the reference case as shown in Fig. 5d.

In the following, we study how $\kappa_{MF}$ variations affect the dominant heat transfer types and temperature distribution with depth. With the $\kappa_{MF}$ decreased from $10^{-13}$ m$^2$ to $10^{-15}$ m$^2$, the number of convection cells is reduced from thirteen to two, and the maximum Darcy velocity of the buoyancy-driven convective flow is decreased from $1.1 \cdot 10^{-7}$ m$\cdot$s$^{-1}$ to $1.4 \cdot 10^{-8}$ m$\cdot$s$^{-1}$ (Figs. 4b and 5b). These variations show that the intensity of the buoyancy-driven convection within the main fault becomes relatively weak and correspondingly, the Pe_MF is also decreased from 9.0 to 0.7 as shown in Fig. 6. Thus, the upward energy

transfer by the buoyancy-driven convection within the main fault is reduced. Conversely, the energy stored at the bottom of the model increases as the $\kappa_{MF}$ is decreased. This is illustrated by the increase in bottom temperature from 432 °C to 519 °C, as shown in Fig. 7.

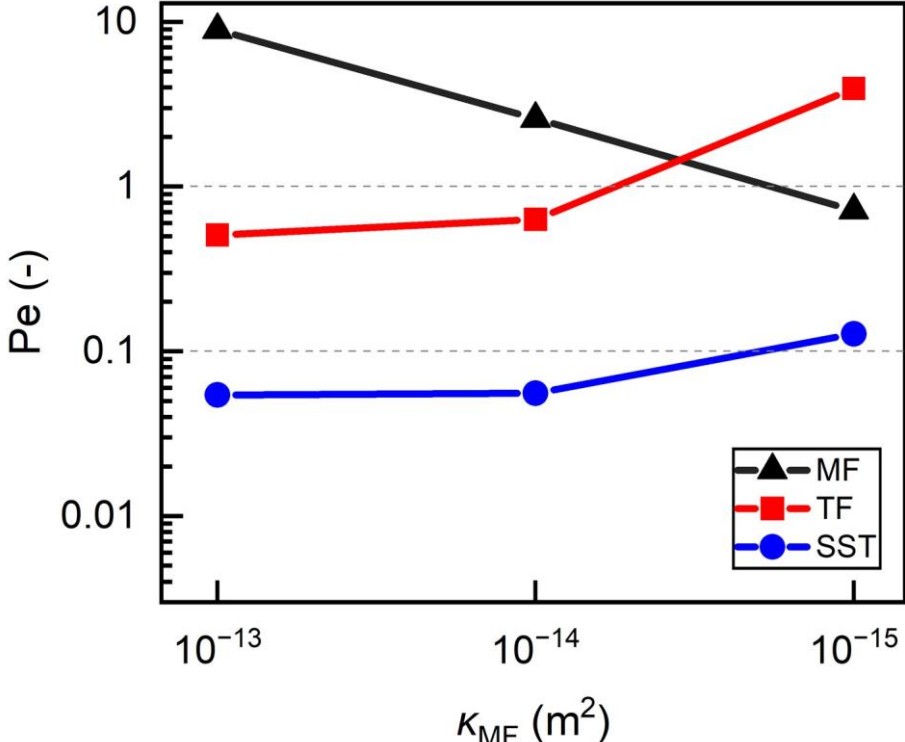

**Figure 6.** Effects of the main fault permeability ($\kappa_{MF}$) on the Pe (median Peclet numbers) of the main fault (MF), transfer faults (TF), and
sandstone (SST).

     Figure 6 shows that for the $\kappa_{MF}$ = $10^{-14}$ m$^2$ case, fluid flow is mainly confined in the faults and the heat transfer is dominated by conduction in the sandstone with Pe_SST < 0.1, in agreement with the reference case. Thus, the energy conducted from the main fault and basement into the sandstone is decreased when the $\kappa_{MF}$ is reduced from $10^{-13}$ m$^2$ to $10^{-14}$ m$^2$. This is evidenced by the decrease in $T_{max}$ from 244 °C to 239 °C as shown by their temperature-depth profiles (i.e., the red and blue

lines in Fig. 7). The $\kappa_{MF}$ = $10^{-15}$ m$^2$ case (i.e., the green line in Fig. 7) results in higher temperature in the range of z = - 12 $\sim$ - 3.5 km than the other two cases. This is due to the larger depth extent (i.e., z = - 9 $\sim$ - 4 km locations) and maximum magnitude (i.e., 130 °C) of the positive thermal anomaly (Fig. 4d) near the central part of the main fault compared to the reference case (Fig. 5c). Furthermore, in the case with $\kappa_{MF}$ = $10^{-15}$ m$^2$, the upward energy transfer within the main fault is less than in the other two cases (i.e., $\kappa_{MF}$ = $10^{-13}$ m$^2$ and $10^{-14}$ m$^2$ cases). However, the enhanced lateral convection and natural advection efficiently

increase the fluid and heat flows from the main fault to the sandstone (with Pe_SST = 0.13) and transfer faults (with Pe_TF = 3.9), higher than the conduction-dominated heat transfer. Consequently, the temperature increases significantly in the sandstone range of z = - 4.9 $\sim$ - 3.9 km as shown by its temperature-depth profile (blue line in Fig. 7), unlike in the other two cases. $\kappa_{MF}$ = $10^{-15}$ m$^2$ results in a higher $T_{max}$ of 285 °C (green line in Fig. 7), which is 41 °C higher than the reference case.

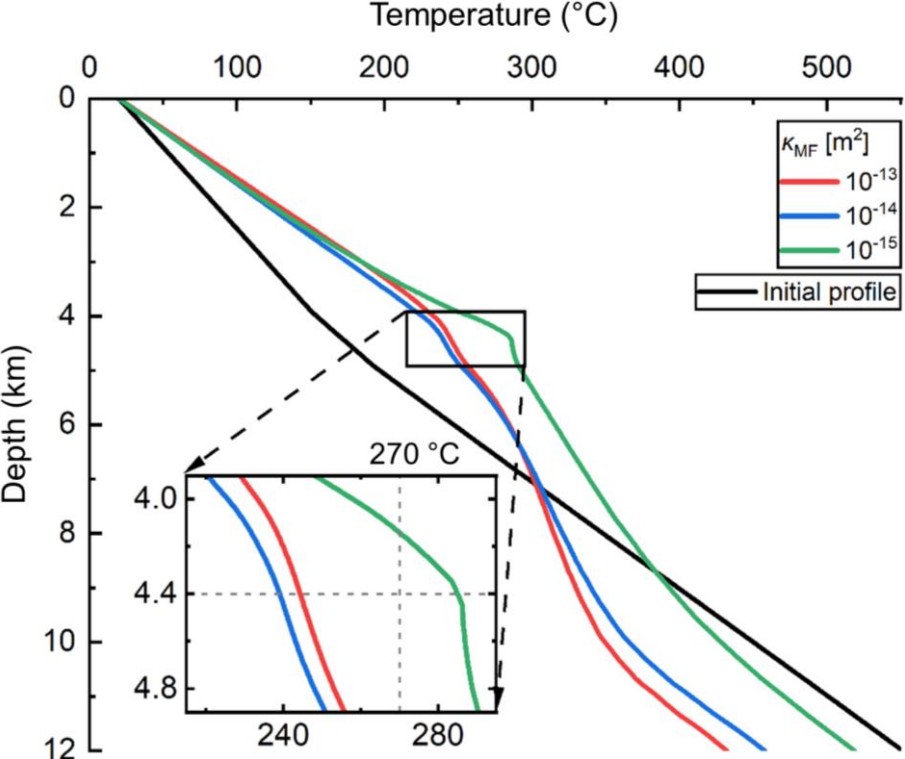

**Figure 7.** Effects of the main fault permeability ($\kappa_{MF}$) on the temperature-depth profiles. The $\kappa_{MF} = 10^{-13}$ m$^2$ case represents the reference case.

### 3.2.2 Sandstone

We investigate the effects of the $\kappa_{SST}$ variations on the transport mechanisms of the hydrothermal convection system. The $\kappa_{SST} = 10^{-16}$ m$^2$ case results in eleven buoyancy-convection cells in the main fault with Pe_MF = 9.1 and a similar maximum

Darcy velocity of the convective flow to the reference case with $\kappa_{SST} = 10^{-15}$ m$^2$. It has a similar concentrated flow in the faults and conduction-dominated heat transfer in the sandstone to the reference case. Results of fluid flow, temperature, and pore pressure change at 1 Myr are shown in Fig. 8 for the $\kappa_{SST} = 10^{-14}$ m$^2$ case. Focusing on Fig. 8b, it is observed in the $\kappa_{MF} = 10^{-14}$ m$^2$ case leads to fourteen convective cells in the main fault with Pe_MF = 8.3 and similar maximum Darcy velocity of the convective flow to the reference case due to their nearly same Pe_MF (Fig. 9).

Initiated by the high $\kappa_{MF}$ and $\kappa_{TF}$, the lateral convection in the sandstone (i.e., Pe_SST = 0.5) and transfer faults (i.e., Pe_TF = 1.0) is limited and consists of fluid recharge and discharge processes as shown in Fig. 8a. The heated fluid recharges from the main fault into the shallow part (i.e., z = -4.4 ~ -3.9 km locations) of the sandstone and transfer faults and discharges from the deep part (i.e., z = - 4.9 ~ - 4.4 km locations) of the sandstone and transfer faults into the main fault, respectively. The maximum Darcy velocity of the recharge and discharge processes reaches 4.6·10$^{-8}$ m·s$^{-1}$, which is ten times higher than

in the reference case (Fig. 8a). Figures 8b and 8c show that the increased mass flow involved in the lateral convection raises the temperature in the sandstone and transfer faults compared to the reference case. The isotherms of mixed concave-convex shapes in the sandstone range demonstrate that the lateral convection results in a higher temperature rise in the shallow part than the deep part of the sandstone and transfer faults. This is due to the energy and fluid losses in the discharge process. From Fig. 8d, it is observed that the lateral convection causes larger (~ -2.9 MPa) and more extensive (~ 1.8 km) pore pressure

disturbances in the sandstone and transfer faults at a depth of 4.4 km than in the reference case.

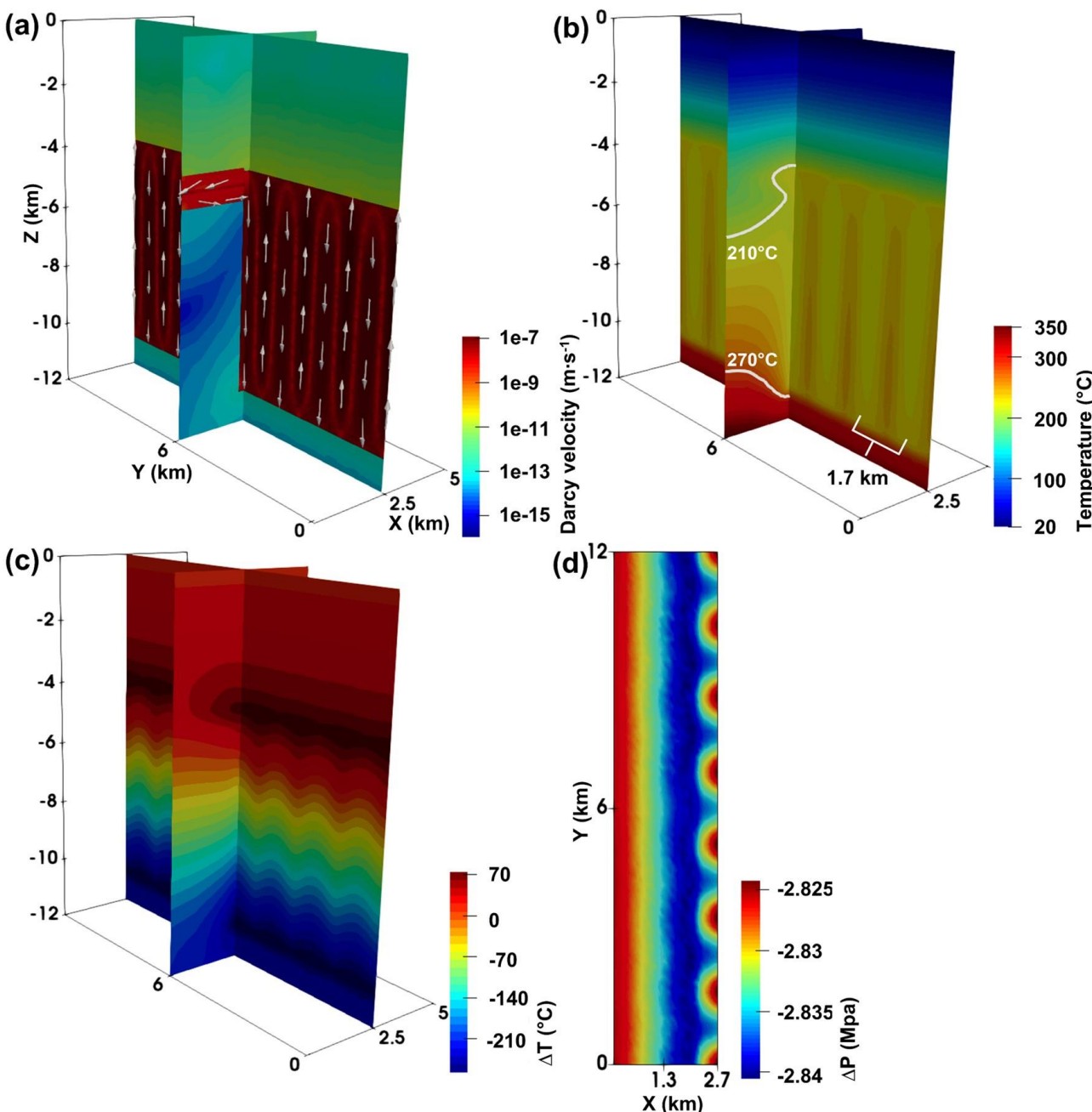

**Figure 8.** Fluid flow, temperature, and pore pressure change fields of the $\kappa_{SST} = 10^{-14}$ case at 1 Myr. Cross-sectional view of the (a) flow field with arrows representing the preferential fluid flow pathways; (b) resulting temperature field. The 210 °C and 270 °C isotherms are highlighted to indicate the dominant heat transfer types; (c) temperature differences compared to the initial conductive temperature distribution; (d) top view of the pore pressure change compared to the initial hydrostatic pore pressure (43.3 MPa) in the western sandstone, transfer faults, and main fault at a depth of 4.4 km. The 1.3 km and 2.7 km along the X-axis follow the west boundary of the Piesberg Quarry and the eastern boundary of the main fault, respectively.

Next, we illustrate how the $\kappa_{SST}$ variations affect the dominant heat transfer types and temperature distribution with depth. With the $\kappa_{SST}$ increased from $10^{-16}$ m$^2$ to $10^{-14}$ m$^2$, the lateral convection in the sandstone and transfer faults is gradually enhanced which is evidenced by the increased Pe_$_{SST}$ from ~ 0 to 0.5 (Fig. 9). This is caused by the reduced pore pressure difference between the main fault and the sandstone due to their decreased permeability contrasts.

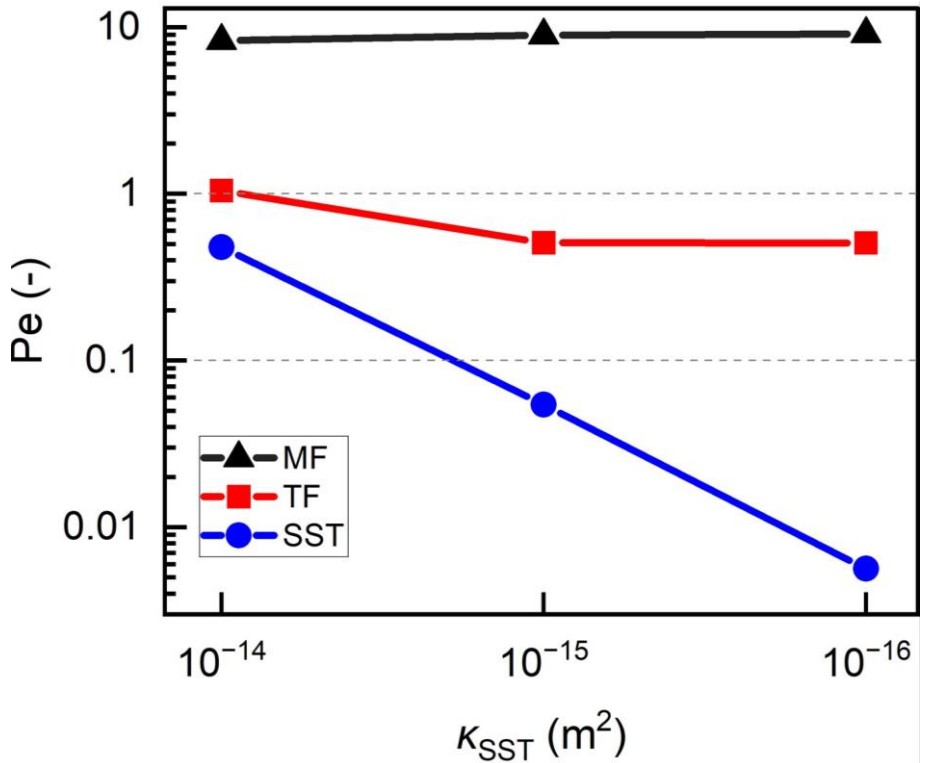

**Figure 9.** Effects of the sandstone permeability ($\kappa_{SST}$) on the Pe (median Peclet numbers) of the main fault (MF), transfer faults (TF), and sandstone (SST).

390       Figure 10 illustrates divergent temperature distributions within the 2 km to 5 km depth for all three $\kappa_{SST}$ cases. The $\kappa_{SST} = 10^{-16}$ m$^2$ case yields a continuous rise of temperature down to 5 km depth due to conduction-dominated heat transfer in the sandstone (Pe_SST≈0) as shown by its temperature-depth profile (i.e., the green line in Fig. 10). A slightly decreased $T_{max}$ (i.e., 244 °C) is observed in the reference case (Fig. 10, red line) compared to the $\kappa_{SST} = 10^{-16}$ m$^2$ case (i.e., 248 °C) (Fig. 10, green line). The reason is that weak lateral convection (Pe_SST = 0.1) transfers energy into the sandstone away from the main

fault and decreases the temperature of the sandstone near the main fault. However, a higher temperature increase within the 2 km to 4.8 km depth is shown for the $\kappa_{SST} = 10^{-14}$ m$^2$ case (Fig. 10, blue line). The $T_{max}$ in the $\kappa_{SST} = 10^{-14}$ m$^2$ case is 269 °C, which is 25 °C higher than in the reference case. This is due to the increased mass flow involved in the enhanced lateral convection (i.e., Pe_SST = 0.5), which transfers the energy from the main fault into the sandstone, transfer faults, and caprock. The energy and fluid losses in the discharge process cause lower temperature rises than the recharge process in lateral

convection. Compared to linear temperature distribution at the initial time, the mixed concave-convex shapes of the temperature profile (Fig. 10, blue line) agree with the isotherms and fluid flow pathways of the lateral convection within the sandstone (Figs. 8a and 8b).

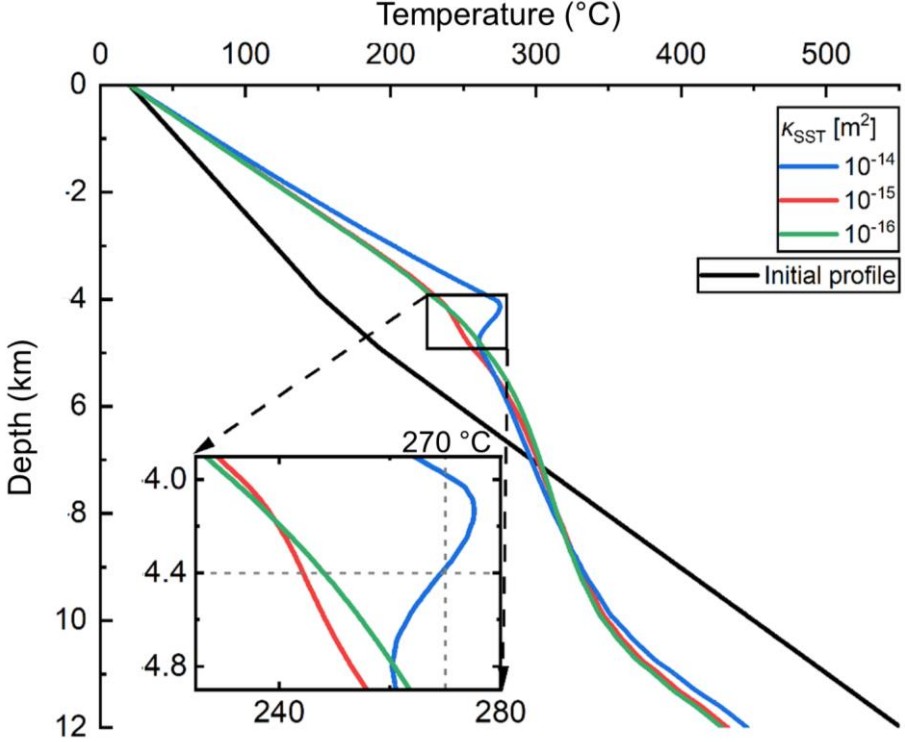

**Figure 10.** Effects of the sandstone permeability ($\kappa_{SST}$) on the temperature-depth profiles. The $\kappa_{SST} = 10^{-15}$ m$^2$ case represents the reference case.

### 3.3    Effects of lateral regional flow

Here we study whether a superimposed LRF changes the fluid flow pathways and types of heat transfer in the hydrothermal convection system. The mutual interaction between LRF and convective/advective flows is also studied. The reference case superimposed by LRFs is used to investigate whether an LRF enhances heat extraction and transfer from the buoyancy-driven convection within the main fault. The results show that forced advection is initiated in the transfer faults only when the LRF exceeds the buoyancy-driven convective flow in the reference case. The dominant heat transfer type in the transfer faults changes from conduction to advection, as proven by the increased Pe_SST from 0.5 to 2.3 (Schilling et al., 2013). The LRF of $5 \cdot 10^{-7}$ m·s$^{-1}$ superimposed on the reference case increases the T$_{max}$ by 8 °C compared to the reference case without LRF (Table 3).

**Table 3.** The effect of maximum lateral regional flow (LRF$_{max}$=$5 \cdot 10^{-7}$ m·s$^{-1}$) on the maximum temperature at the western boundary of the Piesberg Quarry (T$_{max}$) under different permeability distributions of the main fault ($\kappa_{MF}$) and sandstone ($\kappa_{SST}$).

| Permeability distributions (m$^2$) | | T$_{max}$ (°C) | | Effect of LRF$_{max}$ on elevating the T$_{max}$ (°C) |
|---|---|---|---|---|
| | | no LRF | superimposed by LRF$_{max}$ | |
| $\kappa_{MF}$ | $10^{-13}$ | 244 | 252 | 8 |
| | $10^{-14}$ | 239 | 249 | 10 |
| | $10^{-15}$ | 285 | 314 | 29 |
| $\kappa_{SST}$ | $10^{-14}$ | 269 | 271 | 2 |
| | $10^{-15}$ | 244 | 252 | 8 |
| | $10^{-16}$ | 248 | 261 | 13 |

The interactions between the LRF, buoyancy-driven convective flow, and natural advective flow are shown in the $\kappa_{MF}$ variation cases in Table 3. As mentioned in Sect. 3.2.1, the maximum Darcy velocity of the buoyancy-driven convective flow is reduced when $\kappa_{MF}$ is decreased from $10^{-13}$ m$^2$ to $10^{-15}$ m$^2$. Therefore, the extent to which the LRF$_{max}$ overwhelms the buoyancy-driven convective flow increases. This is evidenced by the increased Pe_TF ranging from 2.3 to 6.7. In contrast, the natural advective flow in the transfer faults gets significant when $\kappa_{MF}$ is decreased to $10^{-15}$ m$^2$. Figure 5a shows that the natural advective flow in the western part propagates towards the western boundaries of the model, which is the same as the LRF.

Thus, the LRF$_{max}$ can enhance the heat and mass flow in the transfer faults, leading to a larger increase in T$_{max}$ (ranging from 8 °C to 29 °C) compared to the cases without LRF.

The $\kappa_{SST}$ variation cases in Table 3 illustrate the interactions between the LRF, buoyancy-driven convective flow, and lateral convective flow. Section 3.2.2 shows that the lateral convective flow is gradually weakened (Figs. 4b and 8a) when $\kappa_{SST}$ is decreased from $10^{-14}$ m$^2$ to $10^{-16}$ m$^2$. The maximum Darcy velocity of the buoyancy-driven convective flow is $\sim 1 \cdot 10^{-7}$ m·s$^{-1}$, nearly the same in all $\kappa_{SST}$ cases. Thus, the LRF$_{max}$ can also enhance the heat and mass flows in the sandstone and transfer faults when the $\kappa_{SST}$ is decreased, causing larger T$_{max}$ (from 2 °C to 13 °C) and increased Pe_TF ranging from 2 to 5.4.

According to the above results, the impact of LRF on enhancing the mass and heat flows within the sandstone and transfer faults is reduced for higher $\kappa_{MF}$ and $\kappa_{SST}$ cases (López and Smith, 1996; Sheldon et al., 2012). The reason is that the higher $\kappa_{MF}$ (i.e., $10^{-13}$ m$^2$) and $\kappa_{SST}$ (i.e., $10^{-14}$ m$^2$) lead to a stronger fluid flow and heat transfer in the buoyancy-driven convection and lateral convection, respectively. Thus, the fluid mass fraction that can be impacted by LRF is decreased, resulting in less effect on the increased T$_{max}$ compared to the lower $\kappa_{MF}$ and $\kappa_{SST}$ cases.

## 4    Discussions

### 4.1    Effects of the anisotropic and depth-dependent permeabilities

We consider the anisotropic $\kappa_{MF}$ cases described in Sect. 2.3 to study how the anisotropic permeability affects the transport mechanisms. Preliminary tests show that the cases with the one and three orders degree of anisotropy in the $\kappa_{MF}$ have similar concentrated flow in the faults and conduction-dominated heat transfer in the sandstone to the reference case. They result in a T$_{max}$ of $\sim$ 240 °C, which is 4 °C lower than the reference case. This suggests that the anisotropic main fault acts as a complex pipe-barrier system that favors vertical fluid migration and impedes horizontal fluid migration. In contrast, the $\kappa_{MF}$ case with two orders of magnitude of anisotropy leads to an elevated temperature in the z = - 7 $\sim$ -3 km range, especially in the sandstone range of z = -4.9 $\sim$ -3.9 km, as shown by its temperature-depth profile (i.e., the blue line in Fig. 11). The T$_{max}$ is elevated to 280 °C, which is 36 °C higher than in the reference case. This is caused by the enhanced lateral convection in the sandstone (Pe_SST = 0.1) and natural advection in the transfer faults (Pe_TF = 4.6) that transfer energy more efficiently than the conduction-dominated heat transfer. The lateral convection and natural advection are initiated by the decreased horizontal permeability contrast between the sandstone and main fault, and the increased horizontal permeability contrast between the transfer faults and main fault, respectively.

The depth-dependent $\kappa_{MF}$ cases are used to study depth-dependent permeability effects on transport mechanisms. All the depth-dependent $\kappa_{MF}$ cases have similar focused flow in the faults and conduction-dominated heat transfer in the sandstone to the reference case. However, the number of upward flows within the main fault is decreased in all three cases. When the degree of depth-dependent $\kappa_{MF}$ is reduced from one to three orders of magnitudes, the amount of upward transported energy by the buoyancy-driven convection within the main fault is gradually decreased, resulting in a decreased Pe_MF, ranging between 6.7 and 0.9. The T$_{max}$ is decreased from 217 °C to 195 °C with the decreased degree of depth-dependent $\kappa_{MF}$. Inferred from the in situ lithostatic pressure distribution in the LSB (Manning and Ingebritsen, 1999; Saar and Manga, 2004), the $\kappa_{MF}$ is exponentially decreased by two orders of magnitudes from its top locations to its bottom locations. Its equation is shown as

$$\kappa_{MF}(z) = 10^{-13} \cdot e^{\frac{z+4000}{1520}} \tag{10}$$

where z is the location (m) and $\kappa_{MF}(z)$ is the main fault permeability (m$^2$) at the $z$ (m) location. The resulting temperature profile is shown by its temperature-depth profile (i.e., the green line in Fig. 11) with a T$_{max}$ of 207 °C.

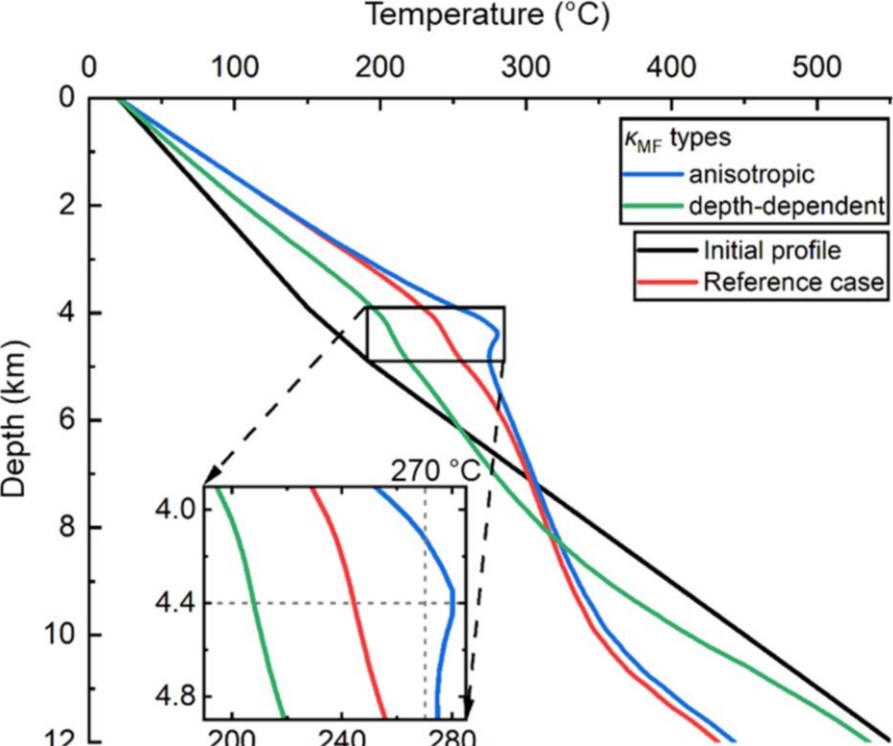

**Figure 11.** Effects of the anisotropic and depth-dependent permeabilities of the main fault ($\kappa_{MF}$) on the temperature-depth profiles.

The anisotropic permeability cases prove that the onset of lateral convection and natural advection relies on the horizontal permeability contrast between the sandstone and main fault, and the horizontal permeability contrast between the transfer faults and main fault, respectively. Influenced by the increasing lithostatic pressure with depth, the mass flow and heat transfer of the buoyancy-driven convection is gradually weakened in the main fault. Thus, the main controlling parameter of the transport mechanism is detailed from the idealized permeability distributions into the spatial permeability distributions of the main fault and sandstone.

### 4.2    Implications for the kilometer-scale thermal anomaly in the Piesberg Quarry and comparable reservoirs

Our study performs the first numerical investigation of possible reasons for the kilometer-scale thermal anomaly ($270 \sim 300$ °C) in the Piesberg Quarry. Since only the geothermometry data from the outcrop samples in the Piesberg Quarry are available, our discussion will be limited to the thermal profiles at 4.4 km depth where the Piesberg Quarry was located during Late Jurassic-Early Cretaceous. The relation between temperatures and mass flows within the sandstone and transfer faults (shown in Figs. 4, 5, 8, 11 and Table 3) suggests that the thermal anomaly results from the coexisting of the buoyancy-driven convection in the main fault, lateral convection in the sandstone rather than from conduction-dominated heat transfer. Both buoyancy-driven convection in the main fault and lateral convection are indispensable to reproducing the measured thermal anomaly. Therein, the buoyancy-driven convection in the main fault is an essential prerequisite for the formation of high temperatures in relatively shallow depths because it can transport heat from deeper to shallow depths (Kohl et al., 2000; Bächler et al., 2003). Moreover, the main role played by lateral convection in the sandstone is to redistribute both heat and fluid mass within the sandstone (Wisian and Blackwell, 2004a), as shown in Fig. 3 and the results obtained in Sect. 3. However, the hypothetical LRF mimicking the effect of topography on the transport mechanisms behave less important than buoyancy-driven convection and lateral convection according to the elevating the $T_{max}$ by LRF (Table 3). Favorable permeability distributions for reproducing the measured thermal anomaly include moderately permeable sandstone (i.e., $10^{-14}$ m$^2$), same permeabilities (i.e., $10^{-15}$ m$^2$) in the main fault and sandstone, and the anisotropic case of the main fault (i.e., $\kappa_{MF\_vertical}/\kappa_{MF\_horizontal} = 10^{-13}$ m$^2$/$10^{-15}$ m$^2$). The $T_{max}$ obtained from the simulations for these three cases are 269 °C, 285 °C, and 280 °C, respectively. These results show that the measured thermal anomaly can be reproduced only if the $\kappa_{MF}$ is equal to or

greater than $10^{-15}$ m$^2$, meanwhile, the ratio of $\kappa_{SST\_horizontal}$ to $\kappa_{MF\ horizontal}$ is equal to or greater than 1. Thus, the magnitude of the $\kappa_{MF}$ and the ratio of $\kappa_{SST\_horizontal}$ to $\kappa_{MF\ horizontal}$ are the determinative factors of the formation of the kilometer-scale thermal anomaly (270 ∼ 300 °C) in the Piesberg Quarry, rather than the LRF. Wuestefeld et al. (2014) and Becker et al. (2019) study the permeability heterogeneity properties in the Piesberg Quarry. Based on the in situ geological and petrophysical properties (i.e., mean permeability distribution, heterogeneities) around Piesberg Quarry and within the LSB (Wuestefeld et al., 2014; Wüstefeld et al., 2017a; Becker et al., 2019), the increased mass flows involved in the coexisting flow modes resulting from the anisotropic case of the main fault (i.e., $\kappa_{MF\_vertical}/\kappa_{MF\_horizontal} = 10^{-13}$ m$^2$/$10^{-15}$ m$^2$), provide the most realistic permeability distribution case to explain the thermal anomaly in the Piesberg Quarry.

In order to make geothermal projects more economically attractive, especially in sedimentary basins (e.g., LSB) with constraint conditions (e.g., low permeability and temperature gradient), fractured/faulted zones are preferentially targeted as valuable geothermal reservoirs (Bruns et al., 2013). As shown in the reference case (Fig. 4), the upward fluid flow within the main fault leads to positive thermal anomalies (i.e., at a maximum of 86 °C) in a relatively shallow depth (∼ 4 km). The relatively shallow locations of geothermal reservoirs greatly decrease the investments (i.e., the costs of drilling and completions) for developing geothermal energy in the sedimentary basin (Huenges and Ledru, 2010). The processes within geothermal reservoirs have important consequences for their successful exploration since the interesting scale at which thermal anomalies could be exploited greatly depend on the heat transfer types and fluid flow pathways (Huenges and Ledru, 2010; Przybycin et al., 2017). If the temperature anomaly is assumed to persist to the present and geologic conditions are constant, for the given reference case geothermal drilling should target the kilometer-scale elevated temperatures (i.e., 2.8 km on the x-axis; 12 km on the y-axis; 2.8 km on the z-axis) around the depth of 4 km (Fig. 4). Our 3D numerical model allows for a better understanding of the processes within geothermal reservoirs (e.g., the role of different selected parameters in the origin, location, scale, and magnitude of thermal anomalies), thus providing tools for predicting and targeting geothermal exploration (Clauser and Villinger, 1990; Guillou-Frottier et al., 2013).

## 5    Conclusions

Inspired by the Piesberg Quarry in northwestern Germany and its established thermal anomaly, we created an idealized numerical model to derive parameters and processes that cause the onset of thermal anomalies in comparable geological settings. The study focused on the developments and mutual interactions of predominant heat transfer types and preferential fluid flow pathways in faulted tight sandstones. The hydraulic parameterizations of the main fault and sandstone are crucial for inferring the mechanisms causing the transport of fluid and heat. Buoyancy-driven convection transports energy upward inside the main fault. Induced by favorable hydraulic conditions (e.g., high sandstone permeability of $10^{-14}$ m$^2$; equal permeability of $10^{-15}$ m$^2$ of the main fault and sandstone), convective flow in the sandstone and advective flow in the transfer faults redistribute both heat and fluid within the reservoir, causing elevated temperatures (≥ 269 °C) and thermal gradients in the sandstones compared to conduction-dominated heat transfer (≤ 250 °C). Lateral regional flow can interact with convective/advective flows in the sandstone and transfer faults and changes the dominant heat transfer from conduction to advection in the transfer faults when it overwhelms the magnitude of the convective flows in the main fault and sandstone. However, its influence is limited by the geometric and hydraulic constraints of the transfer faults and, therefore, significantly reduced compared to the influence of the local permeability distribution. The anisotropic and depth-dependent permeability cases detail the main controlling parameter of the transport mechanism into the spatial permeability distributions of the main fault and sandstone.

The magnitude of the main fault permeability and the ratio of the horizontal permeability of the sandstone to the horizontal permeability of the main fault are the determining factors in the formation of the thermal anomaly at the Piesberg quarry, rather than topographic conditions. The increased mass flows involved in the coexisting flow modes resulting from the anisotropic

case of the main fault (i.e., $\kappa_{MF\_vertical}/\kappa_{MF\_horizontal} = 10^{-13}$ m$^2$/$10^{-15}$ m$^2$), provide the most realistic explanation for the kilometer-scale elevated temperature of 270 ~ 300 °C in the Piesberg Quarry. Our study proves that fractured/faulted zones in tight sandstone reservoirs (a common petroleum reservoir play type) and associated thermal anomalies may be preferred geothermal reservoirs in sedimentary basins worldwide (e.g., LSB). Our 3D numerical model also provides predictive tools for the origin, location, scale, and magnitude of exploitable geothermal reservoirs. More processes (i.e., chemistry and (geo-)mechanics) should be investigated in the further step for the fully coupled transport mechanisms in faulted tight sandstones.

*Code and data availability.* The numerical codes and data are available upon request to the first author.

*Author contributions.* Conceptualization, G.Y., B.B., and R.E.; methodology, G.Y., B.B., and R.E.; software, G.Y., and R.E.; formal analysis, G.Y.; investigation, G.Y.; data curation, G.Y.; writing—original draft preparation, G.Y.; writing—review and editing, B.B., R.E., M.E., K.S., and T.K.; visualization, G.Y.; supervision, T.K. All authors have read and agreed to the published version of the manuscript.

*Conflicts of interest.* The authors declare that they have no conflict of interest.

*Acknowledgments.* This study is part of the subtopic "Geoenergy" in the program "MTET - Materials and Technologies for the Energy Transition" of the Helmholtz Association. The support from the program is gratefully acknowledged. G.Y. is funded by the China Scholarship Council (Grant No. 201709370076) and R.E. is partly by the German Federal Ministry for Economic Affairs and Climate Action in the project INSIDE (Grant No. 03EE4008C). We thank Ali Dashti and Fabian Nitschke (all from Karlsruhe Institute of Technology) for their assistance in reviewing the manuscript as well as Maziar Gholami Korzani (Queensland University of Technology) for his contribution in the early stages of this study. We thankfully acknowledge constructive review comments by Dr. L. Guillou-Frottier and an anonymous reviewer, which improved the quality of this manuscript.

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
