# Peer review of "Transport mechanisms of hydrothermal convection in faulted tight sandstones"

_EGUsphere, 2022_

## Author Comment (AC1)

Dear Editor and Reviewer #1:

We would like to thank Reviewer #1 for your constructive comments concerning our manuscript entitled "Transport mechanisms of hydrothermal convection in faulted tight sandstones" (egusphere-2022-1185). We have addressed each question and comment. The responses to these comments are listed below and the revised manuscript with tracked changes is also submitted.

[0] The authors perform numerical simulation of a geothermal system and the interaction between a deep fault, lateral faults with a sandstone reservoir. The study aims to explain high observed temperatures in the Piesberg quarry, Germany using a sensitivity analysis around the fault and reservoir properties. My comments are mainly about clarification and publication is recommended after a minor revision.

**Answer:** All of Reviewer #1's comments are adopted to improve the scientific quality of our manuscript.

[1] Eq 1: Can you explain what Sm is? I assume it should be based on porosity, fluid density and their compressibilities and that you then can go from a time derivative of mass to a time derivative of pressure. However, is Sm then pressure dependent or assumed constant? Some definitions and assumptions should be given. You also seem to assume single phase.

**Answer:** 1) The "$S_m$" is the constrained specific storage of the porous media (Pa$^{-1}$). The $S_m$ comprised a mechanical alteration in response to pressure and is assumed as a constant given by $S_m = (1-n)/K^s + n/K^l$ with porosity $n$ (-), the bulk moduli of solid $K^s$ and of liquid $K^l$ (Pa) (Watanabe et al. 2017).

2) Yes, we agree with you. Based on assumptions about the constant porosity, temperature/pore pressure-dependent fluid density, and constant compressibility (i.e., reciprocal of the $S_m$) of porous media (Tables 1, 2 and equation (4)), our model can be run from the time derivative of mass to the time derivative of pore pressure.

3) Yes, only single-phase liquid flow is considered in this study.

All the above information has been added to "Chapter 2.2.1 Governing equations".

[2] What is the index m in q$_m$?

**Answer:** The index 'm' has been removed for clarification reasons. The $q$ is the Darcy velocity in the porous media (m·s$^{-1}$).

[3] Eq3 Does 'effective' also include the fluid?

**Answer:** Yes, the 'effective' terms include the fluid and solid phases. All the 'effective' terms have been revised as detailed terms of fluid and solid phases due to clarification reasons.

[4] The density should vary with temperature. Can you comment on how that was modeled?

**Answer:** The fluid density is a temperature/pore pressure-dependent variable. The polynomial function fitting experimental data for pure water in the liquid phase is used to compute fluid density over temperature and pore pressure ranges of 273.15 ~ 1273.15 K and 0 ~ 500 MPa (Linstrom and Mallard 2001). The detailed polynomial function has been added in "Chapter 2.2.2 Numerical model".

[5] 175 Can the authors give the definition of Rayleigh number and its importance? What determines its value?

**Answer:** Linear stability analysis based on Rayleigh number ($Ra$) calculations offer a useful tool to determine the onset of thermal convection (Nield and Bejan 2017). The $Ra$ is a dimensionless number to characterize the fluid's flow regime and is defined as the ratio of buoyancy and viscosity forces multiplied by the ratio of momentum and thermal diffusivities:

$$Ra = \frac{k\left(\rho^l\right)^2 c_{\mathrm{p}}^l \boldsymbol{g} \beta \Delta T H}{\mu[n\lambda^l + (1-n)\lambda^s]}$$

where $k$ is the permeability (m$^2$), $\rho^l$ is the fluid density (kg·m$^{-3}$), $c_p^l$ is the specific heat capacity of fluid (J·m$^{-3}$·K$^{-1}$), $\boldsymbol{g}$ is the gravitational acceleration vector (m·s$^{-2}$), $\beta$ is the coefficient of fluid thermal expansion (K$^{-1}$), $\Delta T$ is the temperature variation (K) over the porous media height $H$ (m), $\mu$ is the fluid dynamic viscosity (Pa·s), $n$ is porosity (-), $\lambda^l$ and $\lambda^s$ are the heat conductivity of the liquid and solid phases (W·m$^{-1}$·K$^{-1}$). The above information has been added to the "Chapter 2.3 Simulation cases".

[6] Table 1: Please add mathematical symbols for identification with the model parameters.

**Answer:** The mathematical symbols of the model parameters have been added in Table 1 for identification.

[7] 188 The kappa values have not been defined as far as I can see. What do they represent? What are the indexes MF and SST? If they correspond to values in table 1, the abbreviations should be listed there as well.

**Answer:** The kappa values had been first defined in Line 155 of "The typical parameterization of main fault permeability ($\kappa_{MF}$) and sandstone permeability ($\kappa_{SST}$) are 10$^{-13}$ m$^2$ and 10$^{-15}$ m$^2$, respectively". The subscript MF means the main fault and the subscript SST represents sandstone. The abbreviations are also added in Table 1.

[8] 205 What is meant by 'initial temperature' in this case? Do you mean based on boundary conditions before flow is accounted for or simply an assumed geothermal gradient? Rather than 'initial temperature', do you not really mean 'initial guess' since you are only interested in the dynamic steady state? A thermal anomaly sounds to me a difference from an expected trend if the circumstances were as everywhere else, e.g. no faults etc. Is that what

you mean?

**Answer:** Yes, we are interested in the dynamic steady state of the hydrothermal convection systems. However, heat conduction dominated the heat transfer in the whole system before the onset of hydrothermal convection. Therefore, the initial temperature is derived from the steady-state simulation of pure heat conduction to avoid the effects of initial temperature perturbation on mass transport and heat transfer. The initial temperature distribution where the fluid flow is not accounted for is dependent on the surface temperature, basal heat flow, heat conductivity, and porosity of porous media.

[9] Can you define mathematically how you define the initial state based on your main equations?

**Answer:** The initial condition of the thermal field is derived from the steady-state simulation of pure heat conduction to avoid the effects of initial temperature perturbation on mass transport and heat transfer. The solution for the steady-state simulation of pure heat conduction in our model is shown as follows:

$$T_{initial}(z) = T_{top} + \frac{Q_{BHF} \cdot z}{n\lambda^l + (1-n)\lambda^s}$$

where $T_{initial}(z)$ is the initial temperature at the depth of $z$ (m), $T_{top}$ is the fixed surface temperature (°C), and $Q_{BHF}$ is the imposed basal heat flow (W·m$^{-2}$). The above information has been added to the "Chapter 2.2.2 Numerical model".

[10] 224 A mathematical definition of Pe number should be given earlier so it is clear how the parameters are combined.

**Answer:** The Peclet number is calculated as the ratio of the heat flow rate by convection to the heat flow rate by conduction for a uniform temperature gradient (Jobmann and Clauser 1994), as follows:

$$Pe = \frac{\rho^l c_p^l \boldsymbol{q} L}{n\lambda^l + (1-n)\lambda^s}$$

where $Pe$ is the Peclet number and $L$ is the length scale of the fluid flow (m). The above information has been added to the "Chapter 2.3 Simulation cases".

[11] In the figures showing temperature vs depth it would be good to include the observed anomaly of the Piesberg quarry one wants to explain since the sensitivity analysis can indicate which parameter is more likely to explain it. For example, figure 7 shows that a high fault permeability is needed to explain a high temperature. Is it within range? Alternatively you could discuss how your model explains the observations more quantitatively in 4.2. Can you state something certain about what is needed for such explanation? If the fault is needed, what properties

must it at least have to realistically explain the observations? Which mechanisms or parameters are less important?

**Answer:** 1) The observed minimum thermal anomaly of 270 °C in the Piesberg quarry has been added to the related sensitivity analysis figures (Figures 4, 7, 10, 11). The zoomed part in these figures shows that temperature-depth profiles located at the right side of the intersection point of the dashed line (i.e., at 4.4 km depth and 270 °C) meet the observed thermal anomaly.

2) Chapter 4.2 has been revised to quantitatively explain the kilometer-scale thermal anomaly. "The observed results show that the measured thermal anomaly can be reproduced only if the $\kappa_{MF}$ is equal to or greater than $10^{-15}$ m$^2$, meanwhile, the ratio of $\kappa_{SST\_horizontal}$ to $\kappa_{MF\ horizontal}$ is equal to or greater than 1. Thus, the magnitude of the $\kappa_{MF}$ and the ratio of $\kappa_{SST\_horizontal}$ to $\kappa_{MF\ horizontal}$ are the determinative factors of the formation of the kilometer-scale thermal anomaly (270 ∼ 300 °C) in the Piesberg Quarry than the lateral regional flow." and other information has been added to the "Chapter 4.2 Implications for the kilometer-scale thermal anomaly in the Piesberg Quarry and comparable reservoirs".

**Reference**

Jobmann, Michael; Clauser, Christoph (1994): Heat advection versus conduction at the KTB: possible reasons for vertical variations in heat-flow density. In Geophysical Journal International 119 (1), pp. 44–68. DOI: 10.1111/j.1365-246X.1994.tb00912.x.

Linstrom, Peter J.; Mallard, William G. (2001): The NIST Chemistry WebBook: A Chemical Data Resource on the Internet. In J. Chem. Eng. Data 46 (5), pp. 1059–1063. DOI: 10.1021/je000236i.

Nield, Donald A.; Bejan, Adrian (2017): Convection in porous media: Springer.

Watanabe, Norihiro; Blöcher, Guido; Cacace, Mauro; Held, Sebastian; Kohl, Thomas (2017): Geoenergy Modeling III. Cham: Springer International Publishing.

---

## Author Comment (AC2)

Dear Editor and Dr. Laurent Guillou-Frottier (Reviewer #2):

Thanks for your valuable and helpful comments concerning our manuscript entitled "Transport mechanisms of hydrothermal convection in faulted tight sandstones" (egusphere-2022-1185). We have addressed your comments carefully and have included the required revisions. The responses to these comments are listed below and the revised manuscript with tracked changes is also submitted.

[0] Review of the manuscript «Transport mechanisms of hydrothermal convection in faulted tight sandstones» by Guoqiang YAN et al., submitted to Solid Earth. December 2022. https://doi.org/10.5194/egusphere-2022-1185.

This manuscript describes the possible fluid flow pathways within a paleo-geothermal system consisting of sandstones, a main fault, and transfer faults. Based on paleotemperatures indicating anomalously high values at a depth of 4.4 km, a numerical model of hydrothermal convection is presented. Unknown parameters (permeability values, anisotropy coefficient) are varied to reproduce the anomalous temperatures. I describe below some important points that have to be addressed. I detail after some minor points. The manuscript should thus be accepted for publication in Solid Earth, after accounting for the suggested revisions that I consider as « minor ».

**Answer:** All of Reviewer #2's comments are adopted to improve the scientific quality of our manuscript.

**[1] Major comments**

[1-1] The adopted methodology is correct, but some pieces of information are missing. The authors used a numerical code for which basic information should be given. In particular, the chosen fluid density law must be specified. Indeed, the authors claim that their benchmark study successfully reproduced the Malkosvky and Magri (2016) experiment (line 176). However, in the Malkosvky and Magri study, a simplified density law (linear approximation) was used, and this approximation may not be valid for temperatures greater than 150°C. If the same simplified density law is used in the entire manuscript, then the results may be strongly biased, all the more that temperatures around 250-300°C are concerned. This point has to be clearly addressed. I suggest the authors to insert an Appendix in which the benchmark experiment would be described and illustrated.

**Answer:** We agree with Reviewer #2's comments on the effect and validity range of fluid density on the development of buoyancy-driven convection and its resulting thermal anomaly. The benchmark experiment of our model has been added in the "Appendix: model validation" to validate against the results of the study conducted by Malkovsky and Magri (2016) and Guillou-Frottier et al. (2020). Different from the fluid density linearly varying with temperature and dynamic viscosity exponentially depending on temperature over the 273.15 ~ 423.15 K range

in the "Appendix: model validation", the specified fluid density and dynamic viscosity law over temperature and pore pressure ranges of 273.15 ~ 1273.15 K and 0 ~ 500 MPa has been added in "Chapter 2.2.2 Numerical model".

[1-2] Another important point deals with the basal thermal condition. The authors impose a basal heat flux of 0.1 W/m², which is much higher than the averaged continental value (67 mW/m²). The updated International Heat Flow database (e.g. Lucazeau, 2019 ; https://doi.org/10.1029/2019GC008389 ) indicates 2 surface heat flow values of 58 and 84 mW/m², at respectively 60 km SW of the area, and 42 km north of the area. There are no other HF values at smaller distances. The closest 100 mW/m² value is located 75 km west of the area (south-east Netherlands). This is not so important since using a lower basal heat flow would simply require increasing the permeability to get the same results, but this choice of 0.1 W/m² must be discussed.

**Answer:** Thanks for Reviewer #2's comments and suggestions on the basal thermal condition (BHF) around the Pieseberg Quarry areas. The reason for the assumed BHF of $0.1$ W·m$^{-2}$ has been properly discussed in "Chapter 2.2.2 Numerical model" referring to the "International Heat Flow database" that Reviewer #2 mentioned.

[1-3] The authors use the Peclet number to qualify the heat transfer regime, which is ok, but what does mean « median Peclet number » ? At what depth is the Peclet number estimated?

**Answer:** The Peclet number is calculated at the scale of finite elements of the model. The heat transfer type and fluid flow pathways are not evenly distributed within the lithological units. Especially, part finite elements of the sandstone and transfer faults close to the main fault always have higher Peclet numbers than the other parts due to the severe convective flow within the main fault. Therefore, the Peclet numbers are not uniform across lithological units. The statistically median values of the Peclet numbers rather than the Peclet number at a specific location (e.g., geometric center) are used to assess the general heat transfer schemes within the lithological units. The above contents have been added in "Chapter 2.3 Simulation cases".

**[2] Minor points**

[2-1] In the Abstract, the depth of 4.4 km should be indicated somewhere, maybe line 14.

**Answer:** Changed as suggested.

[2-2] Line 70: The word "anomaly" suggest that temperatures are greater than an expected value. What is this expected value, or, in other words, at what depth this reservoir was emplaced? The range 270-300 °C may not be anomalous at a depth of 8 km for example.

**Answer:** The "once located at a maximum depth of about 5 km" has been added in Line 70.

[2-3] Line 70: What is the uncertainty of this temperature range? It should depend on the used methodology, but the authors should feed this part with more details on indirect paleotemperature data.

**Answer:** The thermal anomaly with the range of 270 ~ 300 °C of the Piesberg quarry does not mean the uncertain results of the geothermometry data. The reason is that the geothermometry data indicated that the chlorite in veins typically reaches temperatures of ~ 300 °C due to fluid flow, whereas the pore-filling chlorite records an average temperature of 270 °C (Wüstefeld et al. 2017).

[2-4] Figure 1 and caption of Figure 1 are almost identical as in Wüstefeld et al, 2017. Should the authors need an authorization from the Marine and Petroleum Geology Journal?

**Answer:** Figure 1 has been adjusted.

[2-5] Line 106: We have here the answer for the depth of the thermal anomaly. This « 4.4 km » must be indicated before.

**Answer:** The « depth of 4.4 km» have been added in Lines 14 and 70 to state the thermal anomaly's location before Line 106.

[2-6] Line 112. This sentence is too vague. The reader wants to know how the hypothesis of a Lateral Regional Flow has been elaborated. Is there any field observation for this flow? Or is it simply a working hypothesis which will be investigated?

Answer: The "Even though there are no field observations and evidence, lateral regional flow (LRF) resulting from the topography conditions are hypothesized to investigate the effect of topography conditions on the transport mechanisms of hydrothermal convection." has been added before that sentence for clarification reasons.

[2-7] Line 130: add: « and k is permeability (m²) »

**Answer:** Changed as suggested.

[2-8] Line 133 (equation (3)): The equation is wrong: each term of this equation should scale as W/m3, which is not the case in the 3rd term. In addition, radiogenic heat production seems to be neglected, whereas it is mentioned in the definition of basal heat flux (l. 162). If the model was 1 km thick, neglecting heat production should be ok, but we have here a thickness of 12 km, and heat production should probably be accounted for.

**Answer:** 1) The 3$^{rd}$ term ("heat advection term") has been revised and the heat source term $Q$ has been added in equation (3).

2) We agree with Reviewer #2's comment regarding radiogenic heat production. In our model, the heat source

term is considered to be the sum of the mantle heat flow and the heat emitted by the decay of radioactive elements in the crust (Wisian and Blackwell 2004), and it is set at the bottom of the model with a Neumann boundary condition. Kämmlein et al. (2019) show both an enhanced BHF of 0.115 $W \cdot m^{-2}$ and a heat production rate of 6 $\mu W \cdot m^{-3}$ combined with an average BHF of 0.070 $W \cdot m^{-2}$, which can produce an equilibrium temperature log. This means that the high basal heat flow can result in the equivalent effect on the temperature distribution as the combination of the low BHF and additional heat production. For example, in this paper, we used a high BHF value (i.e., 0.1 $W \cdot m^{-2}$) to produce the initial temperature distribution and to represent the total effects of a low BHF (i.e., 0.08 $W \cdot m^{-2}$) and additional radiogenic heat production (i.e., 3.3 $\mu W \cdot m^{-3}$). The BHF of 0.08 $W \cdot m^{-2}$ and radiogenic heat production of 3.3 $\mu W \cdot m^{-3}$ are typical heat flux and radiogenic heat production values in Northern Germany (Achtziger-Zupančič et al. 2017; Fuchs et al. 2022)

[2-9] Figure 2. For clarity, I would also show the main fault in Figure 2b

**Answer:** The main fault has been added in Figure 2(b).

[2-10] Line 167: Topography is not accounted for. Is this hypothesis reasonable and why?

**Answer:** Topography has been considered in the revised version. The "Even though there are no field observations and evidence, lateral regional flows resulting from the topography conditions are hypothesized to investigate the effect of topography on the transport mechanisms of hydrothermal convection." has been added to "Chapter 2.1 The reservoir analog study area".

[2-11] Table 1. « Compressibility »: Where does compressibility appear in the equations? Is it the storage coefficient (same unit and related)?

**Answer:** We are sorry for this mistake. The «Compressibility» and their values have been replaced by the bulk moduli of solid $K^s$ (Pa) in Table 1.

[2-12] Line 175: It would be interesting to detail how the critical Rayleigh number is expressed as a function of the fault geometry, and how this « 61 » number is obtained.

**Answer:** The critical Rayleigh number expressed as a function of the fault geometry has been added, and how this « 61 » number was obtained has been added in Line 175.

[2-13] Line 449: Germany.

**Answer:** Changed as suggested.

[2-14] References: Several references are incomplete (e.g. the first one line 482; but also lines 498, 547, 549,

**Answer:** Changed as suggested.

**References**

Achtziger-Zupančič, P.; Loew, S.; Mariéthoz, G. (2017): A new global database to improve predictions of permeability distribution in crystalline rocks at site scale. In J. Geophys. Res. Solid Earth 122 (5), pp. 3513–3539. DOI: 10.1002/2017JB014106.

Fuchs, Sven; Förster, Andrea; Norden, Ben (2022): Evaluation of the terrestrial heat flow in Germany: A case study for the reassessment of global continental heat-flow data. In Earth-Science Reviews 235, p. 104231. DOI: 10.1016/j.earscirev.2022.104231.

Guillou-Frottier, Laurent; Duwiquet, Hugo; Launay, Gaëtan; Taillefer, Audrey; Roche, Vincent; Link, Gaétan (2020): On the morphology and amplitude of 2D and 3D thermal anomalies induced by buoyancy-driven flow within and around fault zones. In Solid Earth 11 (4), pp. 1571–1595. DOI: 10.5194/se-11-1571-2020.

Kämmlein, Marion; Dietl, Carlo; Stollhofen, Harald (2019): The Franconian Basin thermal anomaly: testing its origin by conceptual 2-D models of deep-seated heat sources covered by low thermal conductivity sediments. In Int J Energy Environ Eng 10 (4), pp. 389–412. DOI: 10.1007/s40095-019-00315-2.

Malkovsky, Victor I.; Magri, Fabien (2016): Thermal convection of temperature-dependent viscous fluids within three-dimensional faulted geothermal systems: Estimation from linear and numerical analyses. In Water Resour. Res. 52 (4), pp. 2855–2867. DOI: 10.1002/2015WR018001.

Wisian, Kenneth W.; Blackwell, David D. (2004): Numerical modeling of Basin and Range geothermal systems. In Geothermics 33 (6), pp. 713–741. DOI: 10.1016/j.geothermics.2004.01.002.

Wüstefeld, Patrick; Hilse, Ulrike; Lüders, Volker; Wemmer, Klaus; Koehrer, Bastian; Hilgers, Christoph (2017): Kilometer-scale fault-related thermal anomalies in tight gas sandstones. In Marine and Petroleum Geology 86, pp. 288–303. DOI: 10.1016/j.marpetgeo.2017.05.015.

---

## Referee Report (RR1)

Review of the REVISED manuscript « *Transport mechanisms of hydrothermal convection in faulted tight sandstones* » by Guoqiang YAN et al., submitted to Solid Earth. February 2023.

My two major points have been addressed in this revised manuscript :

1) An Appendix explaining the benchmark experiment is now included. Fluid density law is discussed and presented in the main text. Everything seems correct.
2) The basal heat flow condition is discussed, lines 220-229, but 0.58 and 0.84 (line 221) have to be changed by 0.058 and 0.084

All other minor points and mistakes have been correctly addressed and the manuscript can be accepted.

L. Guillou-Frottier